# Stiffness and tension gradients of the hair cell's tip-link complex in the mammalian cochlea

Mélanie Tobin[1,2], Atitheb Chaiyasitdhi[1,2†], Vincent Michel[2,3,4†], Nicolas Michalski[2,3,4], Pascal Martin[1,2]*

[1]Laboratoire Physico-Chimie Curie, Institut Curie, PSL Research University, CNRS UMR168, Paris, France; [2]Sorbonne Université, Paris, France; [3]Laboratoire de Génétique et Physiologie de l'Audition, Institut Pasteur, Paris, France; [4]UMRS 1120, Institut National de la Santé et de la Recherche Médicale (INSERM), Paris, France

**Abstract** Sound analysis by the cochlea relies on frequency tuning of mechanosensory hair cells along a tonotopic axis. To clarify the underlying biophysical mechanism, we have investigated the micromechanical properties of the hair cell's mechanoreceptive hair bundle within the apical half of the rat cochlea. We studied both inner and outer hair cells, which send nervous signals to the brain and amplify cochlear vibrations, respectively. We find that tonotopy is associated with gradients of stiffness and resting mechanical tension, with steeper gradients for outer hair cells, emphasizing the division of labor between the two hair-cell types. We demonstrate that tension in the tip links that convey force to the mechano-electrical transduction channels increases at reduced $Ca^{2+}$. Finally, we reveal gradients in stiffness and tension at the level of a single tip link. We conclude that mechanical gradients of the tip-link complex may help specify the characteristic frequency of the hair cell.

DOI: https://doi.org/10.7554/eLife.43473.001

**\*For correspondence:**
pascal.martin@curie.fr

[†]These authors contributed equally to this work

**Competing interests:** The authors declare that no competing interests exist.

## Introduction

The cochlea—the auditory organ of the inner ear—is endowed with a few thousands of mechanosensory hair cells that are each tuned to detect a characteristic sound frequency (*Fettiplace and Kim, 2014*). Different frequencies are detected by different cells, which are spatially distributed in the organ according to a frequency or tonotopic map (*Lewis et al., 1982*; *Greenwood, 1990*; *Viberg and Canlon, 2004*). Despite its critical importance for frequency analysis of complex sound stimuli, determining the mechanism that specifies the characteristic frequency of a given hair cell remains a major challenge of auditory physiology.

Although certainly not the only determinant of hair-cell tuning (*Fettiplace and Fuchs, 1999*), we focus here on the contribution of the hair bundle, the cohesive tuft of cylindrical processes called stereocilia that protrude from the apical surface of each hair cell. The hair bundle works as the mechanical antenna of the hair cell (*Hudspeth, 1989*). Sound evokes hair-bundle deflections, which modulate the extension of elastic elements—the gating springs—connected to mechanosensitive ion channels. By changing the elastic energy stored in the gating springs, sound stimuli affect the channels' open probability, resulting in a mechano-electrical transduction current. Importantly, mechanical stress is conveyed to the mechano-electrical transduction channels by oblique proteinaceous tip links that interconnect the stereocilia near their tips (*Pickles et al., 1984*; *Kazmierczak et al., 2007*). Whether or not the tip link embodies the gating spring, however, is unsure. Electron microscopy (*Kachar et al., 2000*) and molecular dynamics simulations (*Sotomayor et al., 2010*) have suggested that the tip link may be too rigid and thus that a compliant element in series with the tip link could

determine the gating-spring stiffness. If so (but see *Bartsch et al., 2018*), the gating spring could reside within the assemblies of molecules that anchor both sides of the tip link to the actin core of the stereocilia (*Michalski and Petit, 2015*), or perhaps be associated with membrane deformations at the lower insertion point of the tip link (*Powers et al., 2012*). In the following, the molecular assembly comprising the tip link, the transduction channels, as well as the molecules to which they are mechanically connected will be called the 'tip-link complex'.

The operating point of the transducer lies within the steep region of the sigmoidal relation between the transduction current and the hair-bundle position (*Corey and Hudspeth, 1983*; *Russell and Sellick, 1983*; *Johnson et al., 2011*). This key condition for sensitive hearing is thought to be controlled by tension in the tip links at rest (*Hudspeth and Gillespie, 1994*; *Gillespie and Müller, 2009*), as well as by extracellular and intracellular calcium (*Corey and Hudspeth, 1983*; *Ricci et al., 1998*; *Fettiplace and Kim, 2014*), which is thought to stabilize the closed state of the transduction channels (*Hacohen et al., 1989*; *Cheung and Corey, 2006*). Tip-link tension has been estimated at ~8 pN in the bullfrog's sacculus (*Jaramillo and Hudspeth, 1993*) but, to our knowledge, there has been no such report in the mammalian cochlea.

Adaptation continuously resets the mechanosensitive channels to a sensitive operating point when static deflections of the hair bundle threaten to saturate mechanoelectrical transduction (*Eatock, 2000*). Most of the available evidence indicates that movements by molecular motors actively pulling on the tip links and calcium feedback on the open probability of transduction channels contribute to adaptation. With mammalian cochlear hair cells, however, the dependence of adaptation on $Ca^{2+}$ entry has recently been the subject of significant controversy (*Peng et al., 2013*; *Corns et al., 2014*; *Peng et al., 2016*; *Effertz et al., 2017*). Motor forces and calcium feedback can also explain the active hair-bundle movements, including spontaneous oscillations, that have been observed in various species (*Fettiplace and Hackney, 2006*; *Martin, 2008*). Active hair-bundle motility may contribute to hair-cell tuning by actively filtering and amplifying sound inputs (*Hudspeth, 2008*). These findings emphasize the importance of the tip-link complex, including the transduction channels, the tip links that convey sound-evoked forces to these channels, as well as the molecular motors that pull on the tip links, for mechanosensitivity of the hair cell.

Electrophysiological properties of the transduction apparatus, including the activation kinetics and the conductance of the transduction channels, as well as the kinetics of adaptation, have been shown to vary with the characteristic frequency of the hair cell (*Ricci et al., 2003*; *Ricci et al., 2005*; *Fettiplace and Kim, 2014*; *Beurg et al., 2018*). These observations suggest that hair-cell tuning may depend on the transducer itself (*Ricci et al., 2005*). In addition, it is a ubiquitous property of vertebrate auditory organs that the morphology of the hair bundle varies systematically with the characteristic frequency of the corresponding hair cell (*Wright, 1984*; *Lim, 1986*; *Roth and Bruns, 1992*; *Tilney et al., 1992*): going from the high-frequency to the low-frequency region of the organ, the hair bundle gets longer and comprises a progressively smaller number of stereocilia. These morphological gradients have long been recognized as circumstantial evidence that the mechanical properties of the hair bundle might be involved in frequency tuning (*Turner et al., 1981*; *Flock and Strelioff, 1984*; *Fettiplace and Fuchs, 1999*). However, a detailed characterization of mechanical gradients at the level of the whole hair bundle is lacking, in particular to clarify the contribution of the tip-link complex to these gradients.

In this work, we probed passive and active hair-bundle mechanics along the tonotopic axis of an excised preparation of the rat cochlea, within an apical region dedicated to the detection of relatively low sound frequencies for this animal species (1–15 kHz; *Figure 1*). We worked both with inner hair cells, which convey auditory information to the brain and are considered the true sensors of the organ, and with outer hair cells, which are mostly dedicated to cochlear amplification of sound-evoked vibrations (*Hudspeth, 2014*). We combined fluid-jet stimulation to deflect the hair bundle, iontophoresis of a calcium chelator (EDTA) to disrupt the tip links and measure bundle movements resulting from tension released by these links, and patch-clamp recordings of transduction currents to infer the number of intact tip links contributing to the response. From these measurements, we estimated the stiffness of the whole hair bundle, the contribution of the tip links and of the stereociliary pivots to this stiffness, as well as the resting tension in the tip links. Our results reveal mechanical gradients of the tip-link complex according to the tonotopic map and to the division of labor between sensory inner and amplificatory outer hair cells, providing evidence for the implication of the tip-link complex to frequency tuning of cochlear hair cells.

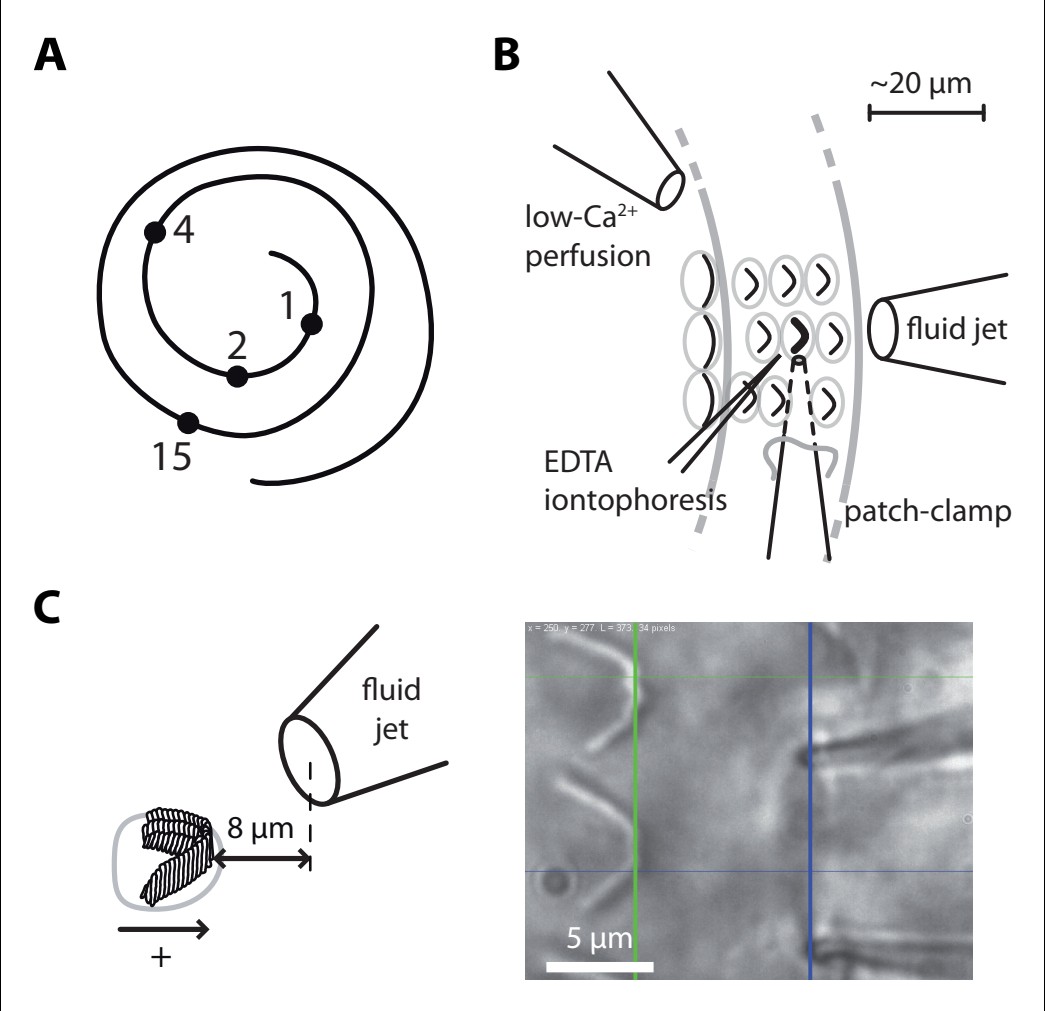

**Figure 1.** Hair-bundle stimulation along the tonotopic axis of the rat cochlea. (**A**) Schematic representation of the tonotopic axis of the rat cochlea. Recordings were made at locations marked by black disks, corresponding to characteristic frequencies (in kHz) increasing from the apex to the base of the cochlea as indicated on the figure and to fractional distances from the apex of 5%, 10%, 20%, and 50%. We report measurements from both inner and outer hair cells at the 1–4 kHz locations, but only from inner hair cells at the 15-kHz location. The rat cochlea was typically 10 mm long. Adapted from *Viberg and Canlon (2004)*. (**B**) Schematic layout of the experimental pipettes around a given outer hair cell. We combined fluid-jet stimulation of single hair bundles, iontophoresis of a $Ca^{2+}$ chelator (EDTA), patch-clamp recordings of transduction currents, and perfusion of low-$Ca^{2+}$ saline. (**C**) Schematic representation of the fluid-jet pipette and of a hair bundle (left) and micrograph of a fluid-jet pipette ready to stimulate an outer hair cell of the rat cochlea (right). A positive (negative) deflection of the hair bundle, as defined on the drawing, was elicited by fluid suction (ejection) into (from) the pipette, promoting opening (closure) of the transduction channels. The horizontal projected distance between the mouth of the pipette (blue vertical line) and the hair bundle (green vertical line) was set at ~8 μm.

DOI: https://doi.org/10.7554/eLife.43473.002

## Results

### The hair-bundle stiffness increases along the tonotopic axis

Using a calibrated fluid jet (Materials and methods; *Figure 2—figure supplements 1–5*), we applied force steps to single hair bundles along the tonotopic axis of the rat cochlea, going from the very apex of the organ to mid-cochlear locations (*Figure 1*). Each position along the tonotopic axis is associated with a characteristic frequency of sound stimulation at which the hair cell is most sensitive in vivo (*Viberg and Canlon, 2004*). For inner hair cells, we probed four positions, corresponding to

characteristic frequencies that varied over nearly four octaves (1–15 kHz). For outer hair cells, we worked at the same three most apical positions as for inner hair cells and the characteristic frequency spanned two octaves (1–4 kHz). Because technical constraints precluded measurements at more basal locations (see Materials and methods), we could only explore a fraction of the tonotopic axis (50% for inner hair cells, 20% for outer hair cells).

Each hair bundle responded to a force step with a fast deflection in the direction of the stimulus followed by a slower movement in the same direction (*Figure 2*; *Figure 2—figure supplement 5A*). Over the duration of the step, the deflection of the hair bundle increased in the direction of the applied step on average by 22% for inner hair cells and by 12% for outer hair cells, corresponding to an apparent softening of the hair bundle by the same amount. A mechanical creep is expected from mechanical relaxation of tip-link tension associated with myosin-based adaptation (*Hudspeth, 2014*). Accordingly, the creep was strongly reduced upon tip-link disruption by EDTA treatment (*Figure 2—figure supplement 5*).

For outer hair cells, we found that a given series of force steps evoked hair-bundle deflections that decreased in magnitude towards more basal locations along the tonotopic axis (*Figure 2A*). Correspondingly, the slope of a bundle's force-displacement relation (*Figure 2B*), and thus stiffness, increased with the characteristic frequency of the hair cell. The same behavior was observed for inner hair cells (*Figure 2C-D*). Remarkably, the stiffness gradient was steeper (*$p<0.05$; *Figure 2—source data 1*) for outer hair cells than for inner hair cells (*Figure 2E*). As the characteristic frequency increased from 1 to 4 kHz, the hair-bundle stiffness $K_{HB}$ increased by 240% from $2.5 \pm 0.2$ mN/m (n = 19) to $8.6 \pm 0.5$ mN/m (n = 21) for outer hair cells, but only by 120% from $1.7 \pm 0.2$ mN/m (n = 19) to $3.8 \pm 0.4$ mN/m (n = 19) for inner hair cells. At the 15-kHz position, where stiffness could only be recorded for inner hair cells (see Materials and methods), $K_{HB} = 5.5 \pm 0.4$ mN/m (n = 14), thus still significantly lower (***$p<0.001$; *Figure 2—source data 1*) than in outer hair cells at the 4-kHz position. At each cochlear position, outer hair-cell bundles were stiffer than inner hair-cell bundles, with a stiffness ratio that increased from the apex to the base of the organ.

As also observed by others using fluid-jet stimulation of cochlear hair cells (*Géléoc et al., 1997*; *Corns et al., 2014*), we measured force-displacement relations that were remarkably linear (*Figure 2B and D*), showing no sign of gating compliance (*Howard and Hudspeth, 1988*). There are at least two possible explanations for this observation. First, the rise time (~500 μs; *Figure 2—figure supplement 3*) of our fluid-jet stimuli may have been too long to outspeed fast adaptation, masking gating compliance (*Kennedy et al., 2003*; *Tinevez et al., 2007*). Second, gating forces – the change in tip-link tension evoked by gating of the transduction channels (*Markin and Hudspeth, 1995*) − may have been too weak to affect hair-bundle mechanics under our experimental conditions (*Fettiplace, 2006*; *Beurg et al., 2008*).

## Parsing out the relative contributions of gating springs and stereociliary pivots to hair-bundle stiffness

There are two contributions to the stiffness of a hair bundle: $K_{HB} = K_{GS} + K_{SP}$. First, hair-bundle deflections modulate the extension of the gating springs that control the open probability of the mechano-electrical transduction channels; we denote by $K_{GS}$ their contribution to hair-bundle stiffness. Second, bending of the actin core of the stereocilia at the stereociliary pivots, as well as stretching horizontal lateral links that interconnect the stereocilia, provides the remaining contribution $K_{SP}$. Because the gating springs are in series with the tip links, disrupting the tip links affords a means to estimate both $K_{GS}$ and $K_{SP}$. We used local iontophoretic application of a $Ca^{2+}$ chelator (EDTA; Materials and methods and *Figure 1*) to disengage the $Ca^{2+}$-dependent adhesion of the cadherin-related molecules forming each tip link (*Kazmierczak et al., 2007*). From the increased magnitude of the hair-bundle response to a given mechanical stimulus (see an example in Figure 4A), we found that the gating springs contributed up to 50% of the total hair-bundle stiffness $K_{HB}$. Averaging over all inner and outer hair cells that we tested, the relative contribution of the gating springs was $r = K_{GS}/K_{HB} = 22 \pm 2$ % (n = 71; *Figure 3—figure supplement 1*), where $1 - r$ is the amplitude ratio of hair-bundle movements before and after tip-link disruption. Both inner and outer hair cells displayed a gradient of gating-spring stiffness $K_{GS} = r\, K_{HB}$ (*Figure 3A*). Between the 1-kHz and the 4-kHz positions, the gating-spring stiffness increased by 520% for outer hair cells but only by 300% for inner hair cells. Similarly, the contribution $K_{SP} = (1 - r)\, K_{HB}$ of the stereociliary

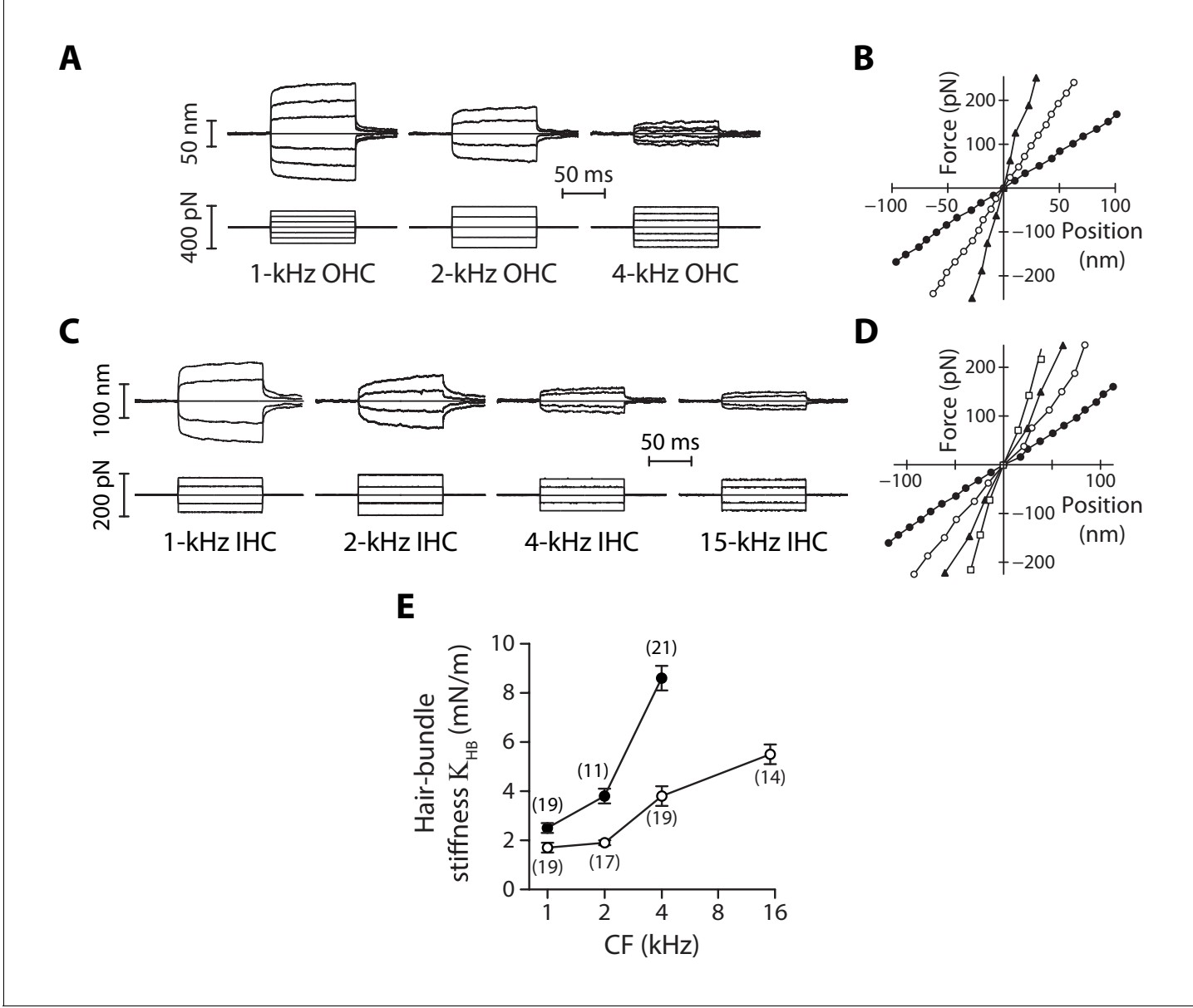

**Figure 2.** Stiffness gradients of the hair bundle. (A) Hair-bundle movements (top) in response to a series of force steps (bottom) for outer hair cells (OHC) with characteristic frequencies of 1, 2, and 4 kHz (from left to right). (B) Force-displacement relations for the data shown in (A), with black disks, white disks and black triangles corresponding to characteristic frequencies of 1, 2, and 4 kHz, respectively. (C) Hair-bundle movements (top) in response to a series of force steps (bottom) for inner hair cells (IHC) with characteristic frequencies of 1, 2, 4, and 15 kHz. (D) Force-displacement relations for the data shown in (C), with black disks, white disks, black triangles, and white squares corresponding to characteristic frequencies of 1, 2, 4, and 15 kHz, respectively. (E) Stiffness ($K_{HB}$) of a hair bundle under control conditions as a function of the characteristic frequency (CF) for inner (white disks) and outer (black disks) hair cells. Each data point in (E) is the mean ± standard error of the mean (SEM) with the number of cells indicated between brackets.
DOI: https://doi.org/10.7554/eLife.43473.003

The following source data and figure supplements are available for figure 2:

**Source data 1.** Statistical significance.
DOI: https://doi.org/10.7554/eLife.43473.009
**Source data 2.** Hair-bundle stiffness of inner and outer hair cells as a function of the characteristic frequency.
DOI: https://doi.org/10.7554/eLife.43473.010
**Figure supplement 1.** Velocity field of a fluid jet.
DOI: https://doi.org/10.7554/eLife.43473.004
**Figure supplement 2.** Geometrical characteristics of a fluid jet.

*Figure 2 continued*

DOI: https://doi.org/10.7554/eLife.43473.005

**Figure supplement 3.** Rise time and linearity of the fluid-jet stimulus.

DOI: https://doi.org/10.7554/eLife.43473.006

**Figure supplement 4.** Test of fluid-jet calibration in the frog's sacculus.

DOI: https://doi.org/10.7554/eLife.43473.007

**Figure supplement 5.** Mechanical creep during a force step.

DOI: https://doi.org/10.7554/eLife.43473.008

pivots to hair-bundle stiffness displayed tonotopic gradients for both inner and outer hair cells (*Figure 3B*).

## Individual gating springs are stiffer in hair cells with higher characteristic frequencies

Both the pivot stiffness $K_{SP}$ and the gating-spring stiffness $K_{GS}$ are expected to vary according to hair-bundle morphology. Hair bundles get shorter and are composed of more numerous stereocilia as one progresses from the apex to the base of the cochlea (*Figure 3—figure supplement 2*), which ought to promote higher stiffness values. Are morphological gradients sufficient to explain the observed stiffness gradients of the hair bundle? Accounting for morphology, we write $K_{SP} = \kappa N_{SP} / h^2$ and $K_{GS} = k_{GS} N_{TL} \gamma^2$, in which $h$, $N_{SP}$, $N_{TL}$ correspond, respectively, to the height, the number of stereocilia and the number of (intact) tip links of the hair bundle, whereas $\gamma \propto 1/h$ is a geometrical projection factor (*Figure 3—figure supplement 2* and *Figure 3—figure supplement 3*). Remarkably, the intrinsic rotational stiffness $\kappa$ of a single stereocilium in outer hair cells remained the same across the positions that we explored (*Figure 3C*; *Figure 3—source data 2*). Similarly, with inner hair cells, there was no significant variation of the rotational stiffness between the 1- and 2-kHz locations as well as between the 4- and 15-kHz locations, although we observed an increase by 100% between the 2- and 4-kHz locations (*Figure 3—source data 2*). Averaging over the ensembles of outer and inner hair cells that we probed, the rotational stiffness $\kappa = 1.2 \pm 0.2$ fN·m/rad (n = 79) in outer hair cells was about 140% higher than the value $\kappa = 0.5 \pm 0.1$ fN·m/rad (n = 137) measured in inner hair cells. In contrast, the intrinsic stiffness $k_{GS}$ of a single gating spring increased by 180% from $1.3 \pm 0.4$ mN/m (n = 29) to $3.7 \pm 0.7$ mN/m (n = 17) in outer hair cells and by 240% from $0.5 \pm 0.2$ mN/m (n = 14) to $1.7 \pm 0.3$ mN/m (n = 30) in inner hair cells, for characteristic frequencies that increased from 1 to 4 kHz and from 1 to 15 kHz, respectively (*Figure 3D*). Thus, morphological gradients can account for the observed gradient in pivot stiffness $K_{SP}$, but not for the observed gradient in gating-spring stiffness $K_{GS}$ and in turn for the whole hair-bundle stiffness $K_{HB}$. The hair-bundle morphology is not the sole determinant of hair-bundle mechanics.

## Tip-link tension increases along the tonotopic axis

We then estimated the mechanical tension in the tip links at rest, that is in the absence of an external stimulus. The transduction channels close when the tip links are disrupted, indicating that the channels are inherently more stable in a closed state (*Assad et al., 1991*; *Beurg et al., 2008*; *Indzhykulian et al., 2013*). In functional hair bundles, tip-link tension is thought to bring the operating point of the transducer within the steep region of the sigmoidal relation between the channels' open probability and the position of the hair bundle, ensuring sensitive detection of hair-bundle deflections. If there is tension in the tip links, then disrupting these links must result in a positive offset in the resting position of the hair bundle (*Assad et al., 1991*; *Jaramillo and Hudspeth, 1993*).

In response to iontophoresis of a $Ca^{2+}$ chelator (EDTA), we observed a net positive movement $\Delta X_R$ of the hair bundle at steady state, as expected if the tip links broke and released tension (*Figure 4A*). Consistent with tip-link disruption, this movement was associated with a decrease in hair-bundle stiffness, as well as with closure of the transduction channels and loss of transduction (*Figure 4B*). The positive offset in resting position upon tip-link disruption was observed at all positions that we explored along the tonotopic axis of the cochlea, both for inner and outer hair cells, demonstrating that the hair bundles were indeed under tension (*Figure 5A*). In addition, we observed that the magnitude of the evoked movement increased significantly (\*\*p<0.01; *Figure 5—*

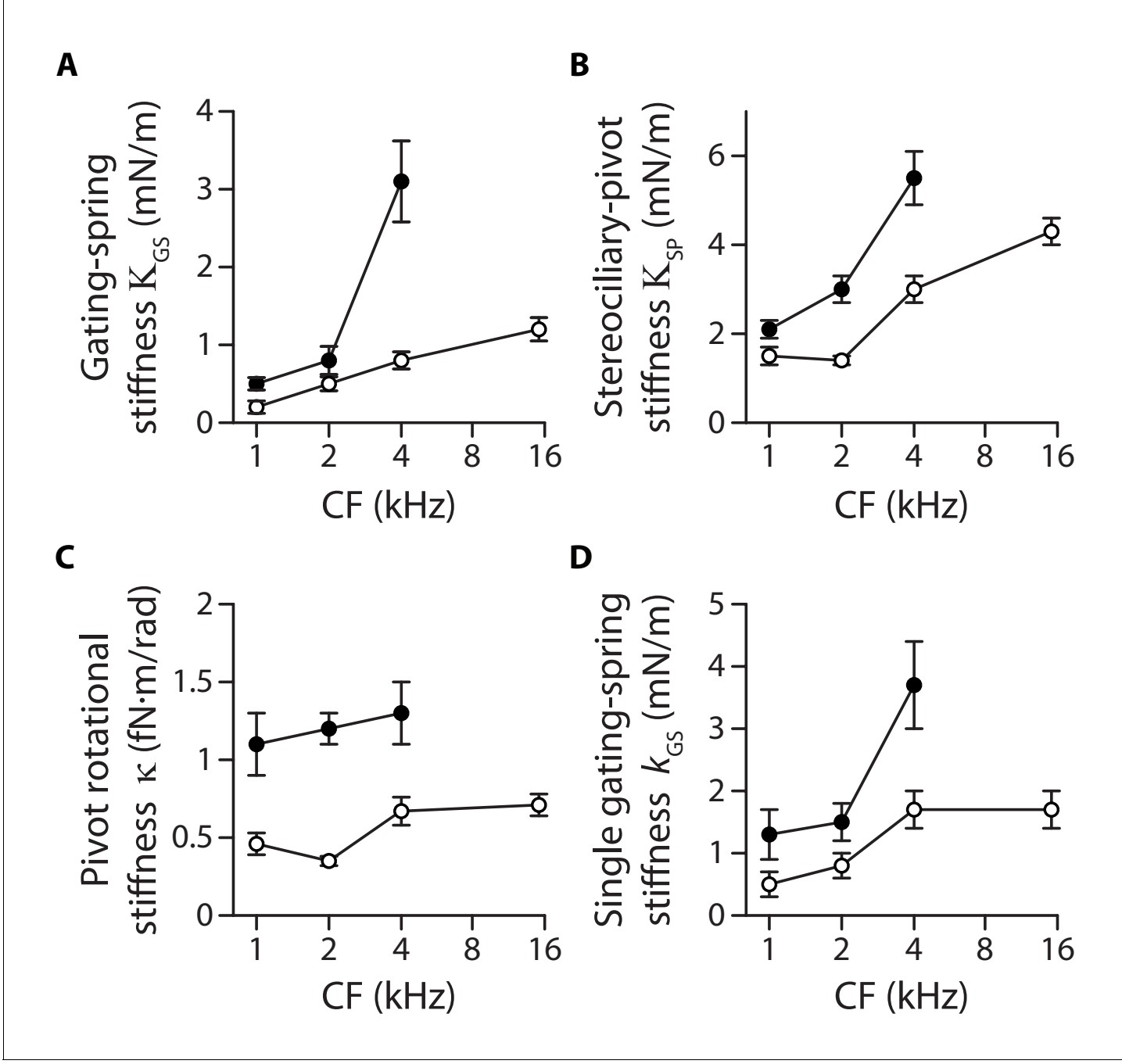

**Figure 3.** Stiffness gradients of the gating springs and of the stereociliary pivots. Stiffness (**A**) of the contribution of the gating springs ($K_{GS} = r\,K_{HB}$), (**B**) of a hair bundle after tip-link disruption, corresponding to the contribution of the stereociliary pivots ($K_{SP} = (1 - r)\,K_{HB}$), (**C**) of a single stereociliary pivot ($\kappa = K_{SP}\,h^2/N_{SP}$), and (**D**) of a single gating spring ($k_{GS} = K_{GS} / (\gamma^2 N_{TL})$) as a function of the characteristic frequency (CF) for inner (white disks) and outer (black disks) hair cells. These stiffnesses were calculated from measured values of the hair-bundle stiffness $K_{HB}$ (**Figure 2**), the amplitude ratio $1 - r$ of hair-bundle movements before and after tip-link disruption (**Figure 3—figure supplement 1**), the hair-bundle height $h$ and the number of stereocilia $N_{SP}$ (**Figure 3—figure supplement 2**), and the average number $N_{TL}$ of intact tip links (**Figure 3—figure supplement 3**). Each data point is the mean ± SEM; SEMs were calculated as described in the Materials and methods.

DOI: https://doi.org/10.7554/eLife.43473.011

The following source data and figure supplements are available for figure 3:

**Source data 1.** Morphological parameters of inner and outer hair-cell bundles.

DOI: https://doi.org/10.7554/eLife.43473.015

**Source data 2.** Statistical significance.

*Figure 3 continued on next page*

*Figure 3 continued*

DOI: https://doi.org/10.7554/eLife.43473.016

**Source data 3.** Gating-spring contribution to the hair-bundle stiffness.

DOI: https://doi.org/10.7554/eLife.43473.017

**Source data 4.** Hair-bundle morphology along the tonotopic axis.

DOI: https://doi.org/10.7554/eLife.43473.018

**Source data 5.** Transduction currents and number of intact tip links along the tonotopic axis.

DOI: https://doi.org/10.7554/eLife.43473.019

**Figure supplement 1.** Gating-spring contribution to the hair-bundle stiffness.

DOI: https://doi.org/10.7554/eLife.43473.012

**Figure supplement 2.** Hair-bundle morphology along the tonotopic axis.

DOI: https://doi.org/10.7554/eLife.43473.013

**Figure supplement 3.** Transduction currents and number of intact tip links along the tonotopic axis.

DOI: https://doi.org/10.7554/eLife.43473.014

*source data 1*) from $9 \pm 3$ nm (n = 13) to $45 \pm 10$ nm (n = 12) for outer hair cells with characteristic frequencies that increased from 1 to 4 kHz. In contrast, we observed no significant difference among inner hair cells with characteristic frequencies that varied within the 1–15-kHz range (p>0.05; *Figure 5—source data 1*): the positive offset was $21 \pm 2$ nm (n = 71) over the whole ensemble of inner hair cells.

As a result, within the range of cochlear locations that we explored, we measured a steep gradient of hair-bundle tension for outer hair cells but a comparatively weaker gradient (*p<0.05; *Figure 5—source data 1*) for inner hair cells (*Figure 5B*). Tension $T_R = K_{SP} \, \Delta X_R$ in the hair bundle was estimated as the product of the pivot stiffness $K_{SP}$ and the positive offset $\Delta X_R$ in resting position evoked by tip-link disruption (Materials and methods). The hair-bundle tension increased by nearly 13-fold from $18 \pm 7$ pN (n = 15) to $248 \pm 59$ pN (n = 17) for outer hair cells (characteristic frequencies: $1-4$ kHz) but by only 3.3-fold from $24 \pm 5$ pN (n = 33) to $100 \pm 22$ pN (n = 12) for inner hair cells (characteristic frequencies: $1-15$ kHz). Tension in the hair bundle resulted from the summed contributions of tension in individual tip links. Dividing the tension $T_R$ by the average number $N_{TL}$ of intact tip links in our recordings and projecting the result along the oblique axis of the tip links (projection factor $\gamma$) provided estimates of the tension $t_R = T_R / (\gamma \, N_{TL})$ in a single tip link. Remarkably, the observed gradients in hair-bundle tension (*Figure 5B*) were not only due to an increase in the number of tip links that contributed to this tension (*Figure 3—figure supplement 3*), for tension in a single tip link also showed gradients (*Figure 5C*). The single tip-link tension was comparable in the two types of cells at the 1-kHz location: $4.7 \pm 2.0$ pN (n = 23) for outer hair cells and $5.9 \pm 1.3$ pN (n = 43) for inner hair cells. However, at the 4-kHz location, the single tip-link tension had increased by 620% to $34 \pm 8$ pN (n = 19) in outer hair cells but only by 170% to $16 \pm 4$ pN (n = 17) in inner hair cells; at the 15-kHz location, tip-link tension in inner hair cells was $19 \pm 5$ pN (n = 18). A linear regression of the relation between the single tip-link tension and the characteristic frequency confirmed that the gradient was significantly (*p<0.05; *Figure 5—source data 1*) steeper for outer hair cells.

Note that we have estimated here the stiffness and tension of a single tip-link complex along the hair bundle's axis of bilateral symmetry, which corresponds to the natural axis of mechanical stimulation in the cochlea. If the horizontal projection of tip links was actually oriented at an angle with respect to this axis, stiffness and tension ought to be larger along the tip-link axis than the values reported here. A systematic change in tip-link orientation along the tonotopic axis would also affect the gradients. These effects are expected to be small in inner hair cells, which are endowed with nearly linear hair bundles, but could be relevant for the V-shaped hair bundles of outer hair cells (*Figure 3—figure supplement 2* and *Pickles et al., 1987*).

## Tip-link tension first increases upon $Ca^{2+}$ chelation

The dynamic response to an iontophoretic step of EDTA, and thus to a decrease of the extracellular $Ca^{2+}$ concentration, was biphasic. The hair bundle first moved in the negative direction (arrowhead in *Figure 4A*), before the directionality of the movement reverted and the bundle showed the positive movement associated with tip-link disruption. The negative movement was associated with an increased inward current of similar time course (*Figure 4B*). Within the framework of the gating-

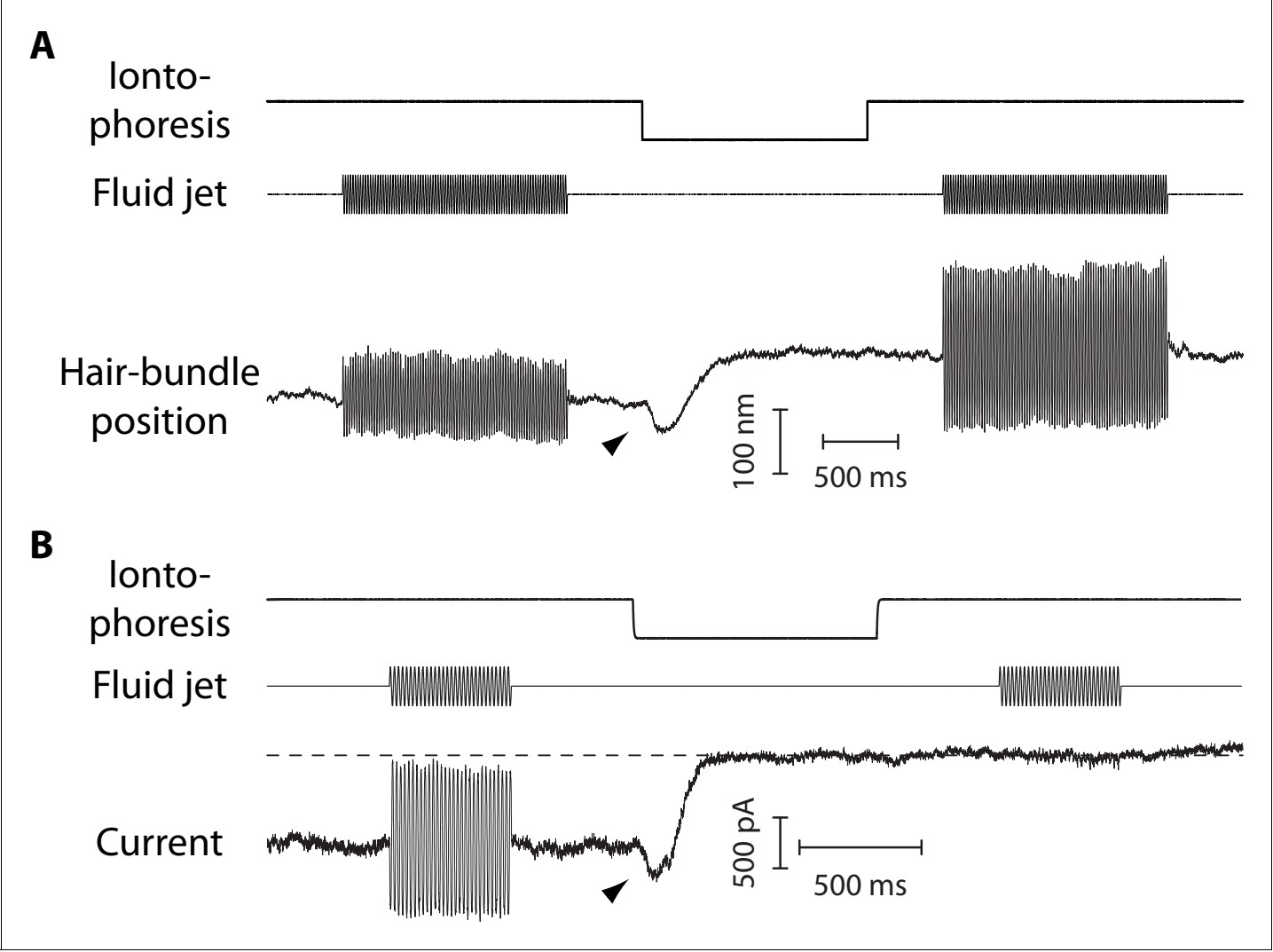

**Figure 4.** Mechanical and electrical response of a hair bundle to fluid-jet stimulation and fast calcium chelation. (A) An iontophoretic step of a calcium chelator (EDTA; top) elicited a biphasic movement of the hair bundle from an inner hair cell (bottom): the hair bundle first moved in the negative direction (arrowhead) and then in the positive direction. After iontophoresis, the position baseline was offset by $\Delta X_R = +78$ nm with respect to the resting position at the start of the experiment. A sinusoidal command to a fluid jet (middle) evoked hair-bundle movements (bottom) that increased in magnitude, here by 50%, after application of the iontophoretic step. Repeating the iontophoretic step elicited no further movement and the response to fluid-jet stimulation remained of the same magnitude. A similar behavior was observed with 101 inner and 44 outer hair cells. (B) An iontophoretic step of EDTA (top) also elicited biphasic variations of the transduction current: the inward current first increased (arrowhead) and then decreased. Before application of the calcium chelator, fluid-jet stimulation evoked a transduction current of 1.5-nA peak-to-peak magnitude; the open probability of the transduction channels was near ½. The transduction current was abolished by the iontophoretic step. Outer hair cell at the 4-kHz location; the same behaviour was observed with 17 outer hair cells. In (A–B), the command signal to the fluid-jet device was a 60-Hz sinusoid and we applied a −100-nA iontophoretic step on top of a +10-nA holding current. The hair bundles were exposed to 20-µM $Ca^{2+}$. In (B), the dashed line indicates the current for which the transduction channels are all closed.

DOI: https://doi.org/10.7554/eLife.43473.020

spring model of mechanoelectrical transduction (*Corey and Hudspeth, 1983*; *Markin and Hudspeth, 1995*), this observation is readily explained if the evoked decrease in the extracellular $Ca^{2+}$ concentration resulted in an increase in gating-spring tension, which both pulled the hair bundle in the negative direction and led to the opening of the transduction channels.

The magnitude of the negative movement at the peak showed no significant gradient and was similar between inner and outer hair cells, with an average magnitude of $\Delta X_{Ca} = -26 \pm 2$ nm over the whole ensemble of hair cells (n = 83; *Figure 6A*). However, because morphological gradients

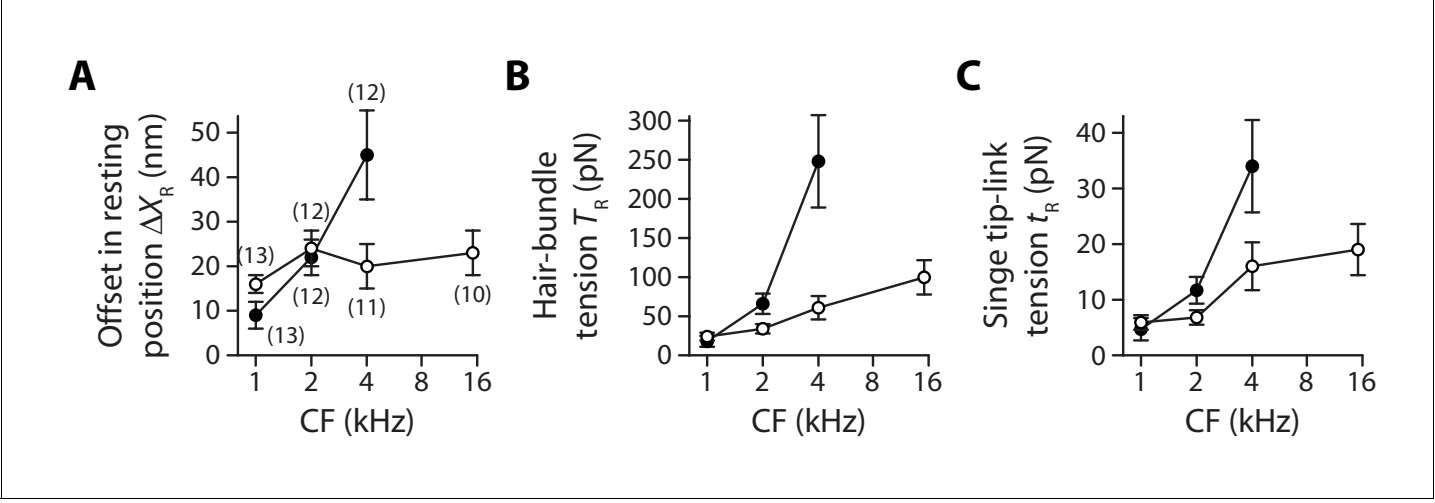

**Figure 5.** Gradients in tip-link tension at rest. Offset $\Delta X_R$ in the resting position of a hair bundle resulting from tension release in the tip links (**A**), tip-link tension $T_R = K_{SP} \, \Delta X_R$ in the hair bundle (**B**) and tension $t_R = T_R / (\gamma \, N_{TL})$ along the oblique axis of a single tip link (**C**) as a function of the characteristic frequency (CF) for inner (white disks) and outer (black disks) hair cells. The hair-bundle tension $T_R$ (**B**) was calculated as the product of the stereociliary-pivot stiffness $K_{SP}$ shown in **Figure 3** and the data shown in (**A**); this tension is estimated along the bundle's horizontal axis of mirror symmetry. The single tip-link tension $t_R$ was then deduced from the projection factor $\gamma$ and the average number $N_{TL}$ of intact tip links in a hair bundle (**Figure 3—figure supplement 3**). Each data point in (**A**) is the mean ± SEM with the number of cells indicated between brackets; in (**B–C**), mean values and SEMs were calculated as described in the Materials and methods.
DOI: https://doi.org/10.7554/eLife.43473.021

The following source data is available for figure 5:

**Source data 1.** Statistical significance.
DOI: https://doi.org/10.7554/eLife.43473.022

**Source data 2.** Offset in the resting position of a hair bundle upon tip-link disruption.
DOI: https://doi.org/10.7554/eLife.43473.023

(**Figure 3—figure supplement 2**) resulted in gradients of pivot stiffness $K_{SP}$ (**Figure 3B**), the maximal increase $\Delta T = -K_{SP} \, \Delta X_{Ca}$ in hair-bundle tension was larger for hair cells with higher characteristic frequencies (**Figure 6B**), as was the maximal tension $t_{max}$ that a single tip link sustained before tip-link disruption (**Figure 6C**). Going from the 1-kHz location to the 4-kHz location, this maximal tip-link tension displayed a gradient from 14 ± 4 pN (n = 30) to 54 ± 12 pN (n = 23) in outer hair cells and from 15 ± 3 pN (n = 49) to 44 ± 8 pN (n = 39) in inner hair cells; the maximal tension in inner hair cells at the 15-kHz location was not significantly different than at the 4-kHz location.

When immersing the hair cells in low-$Ca^{2+}$ saline, the negative movement was always followed by tip-link disruption and could thus not be observed twice with the same hair bundle. However, in six different preparations for which the hair bundle was immersed in saline with a higher $Ca^{2+}$ concentration (500 μM) than usual (20 μM), we were able to preserve the integrity of the tip links and demonstrate that the negative movements could be reversible (**Figure 6D**). Under such conditions, we observed that the absolute magnitude and the speed of the negative movement increased with the magnitude of the iontophoretic current. Notably, the hair bundle reached a new steady-state position when the iontophoretic step was long enough (**Figure 6E**), suggesting that resting tension in the tip links could be modulated by the extracellular $Ca^{2+}$ concentration, with higher tensions at lower $Ca^{2+}$ concentrations.

## Discussion

Tonotopy of the mammalian cochlea is known to be associated with gradients of hair-bundle morphology (**Wright, 1984**; **Lim, 1986**; **Roth and Bruns, 1992**; **Tilney et al., 1992**), as well as of electrophysiological properties of the transduction apparatus (**Ricci et al., 2003**; **Ricci et al., 2005**; **Fettiplace and Kim, 2014**; **Beurg et al., 2018**). The work presented here reveals that tonotopy is

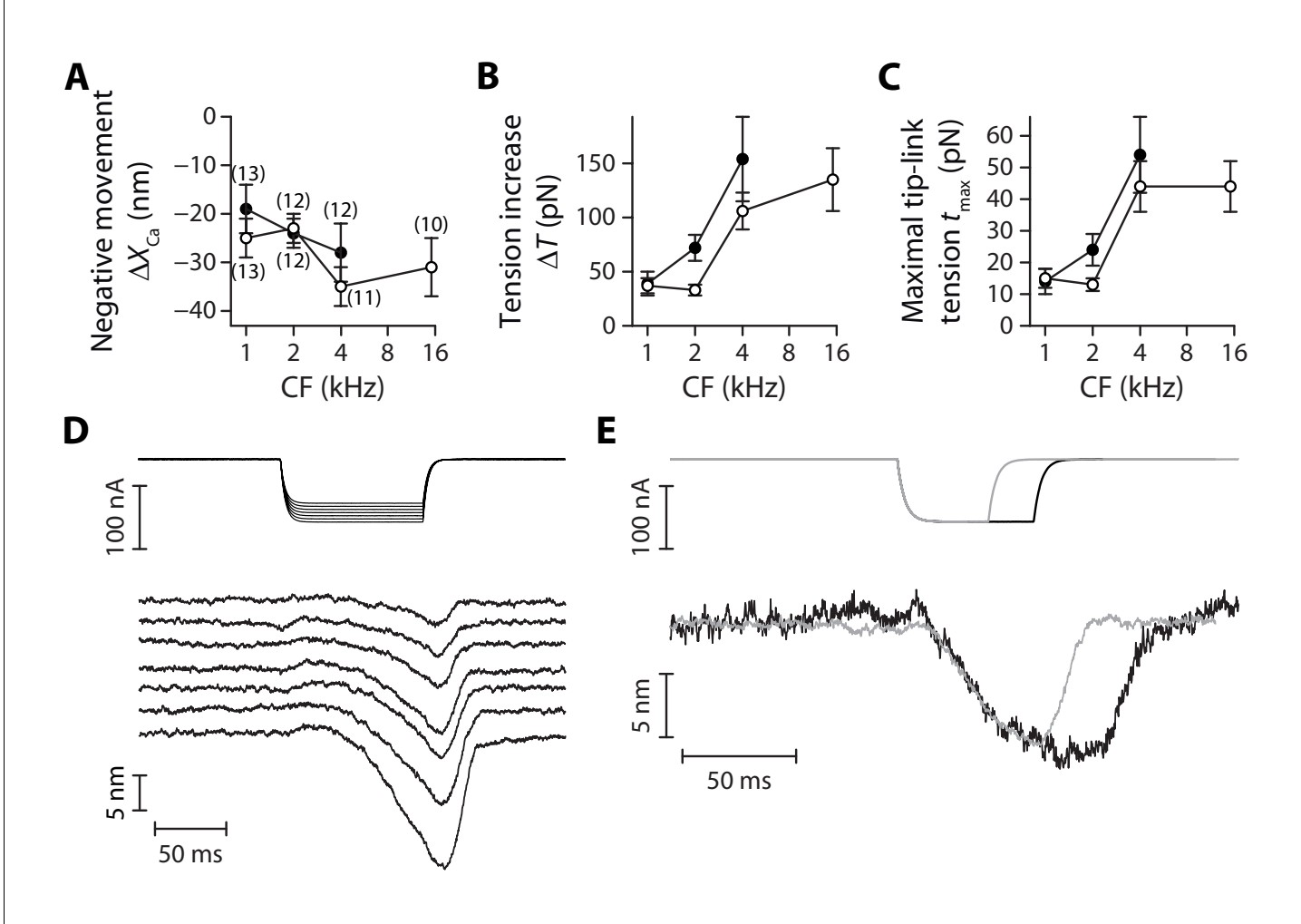

**Figure 6.** Tensioning of the tip links at decreased $Ca^{2+}$ concentrations. The amplitude of the negative hair-bundle movement $\Delta X_{Ca}$ (A), of the maximal increase $\Delta T = -K_{SP}\,\Delta X_{Ca}$ in hair-bundle tension (B), and of the maximal tension $t_{max} = t_R + \Delta T/(\gamma\,N_{TL})$ in a single tip link (C) are plotted as a function of the hair cell's characteristic frequency (CF). The tension increase in (B) was calculated from the stiffness $K_{SP}$ of the stereociliary pivots (*Figure 3B*) and the data shown in (A). The single tip-link tension $t_{max}$ was then deduced from the tension at rest $t_R$ in a single tip link (*Figure 5c*), the projection factor $\gamma$ (*Figure 3—figure supplement 2*) and the average number $N_{TL}$ of intact tip links (*Figure 3—figure supplement 3*). (D) Current-step commands (top) applied to an iontophoretic pipette containing the $Ca^{2+}$ chelator EDTA evoked reversible negative movements of the hair bundle (bottom). (E) When the stimulus (top) was long enough, the hair bundle position could reach a steady state (bottom), corresponding to higher resting tension in the tip links. In (A–C), the hair bundles were immersed in low-$Ca^{2+}$ saline, for which EDTA iontophoresis led to tip-link disruption. Positions and tensions were estimated at the point of polarity reversal of the hair-bundle movement (see *Figure 4A*), thus at the initiation of tip-link disruption, where the hair bundle reached its largest deflection in the negative direction and tension was thus maximal. Black and white disks correspond to outer and inner hair cells, respectively. Each data point in (A) is the mean ± SEM with numbers of cells indicated between brackets; in (B–C), mean values and SEMs were calculated as described in the Materials and methods. In (D–E), the hair bundles were immersed in a saline containing 500-μM $Ca^{2+}$; this higher $Ca^{2+}$ concentration preserved the integrity of the tip links upon EDTA iontophoresis.

DOI: https://doi.org/10.7554/eLife.43473.024

The following source data is available for figure 6:

**Source data 1.** Statistical significance.
DOI: https://doi.org/10.7554/eLife.43473.025
**Source data 2.** Negative movement $\Delta X_{Ca}$ of the hair-bundle before tip-link disruption.
DOI: https://doi.org/10.7554/eLife.43473.026

also associated with gradients of intrinsic mechanical properties of the hair cell's tip-link complex. Specifically, by dissecting the relative contributions of the tip links and of the stereociliary pivots to the micromechanical properties of the hair bundle, we found that the gating springs that control the open probability of the mechanoelectrical transduction channels are stiffer (*Figure 3D*) and subjected to higher mechanical tension (*Figure 5C*) in hair cells that respond to higher characteristic frequencies. In return, our data raises the possibility of a mechanical role of the tip-link complex in the process that sets the characteristic frequency of the hair cell, at least within the apical cochlear region that we probed in this study. Whether or not stiffness and tension of the tip-link complex continue increasing toward more basal locations remains to be determined.

The stiffness $K_{HB}$ of the whole hair bundle displayed steeper gradients than those expected using the rough estimate $K_{HB} \propto N_{SP}/h^2$ from morphological changes (*Figure 3—figure supplement 2*) in height $h$ and number of stereocilia $N_{SP}$. Computing the stiffness ratio between the most basal and the most apical cochlear location that we were able to probe, the measured stiffness ratios (*Figure 2E*) were ~50% and ~70% larger than those expected from morphology for outer and inner hair cells, respectively. We interpret this result as the consequence of intrinsic gradients of the single gating-spring stiffness (*Figure 3D*). Further emphasizing mechanical regulation at the level of the tip-link complex, we also observed that the rotational stiffness of a single stereocilium was nearly uniform across the cochlear locations that we tested, especially in outer hair cells (*Figure 3C*). Stiffness gradients of hair bundles with disrupted tip links are thus entirely determined by morphology, in contradistinction to those observed with intact hair bundles.

## Role of hair-bundle maturation in mechanical gradients

Our experiments were performed with hair cells from juvenile animals (P7-P10), before the onset of hearing. Hair-cell maturation progresses from base to apex in the cochlea (*Wu and Kelley, 2012*), which may thus have affected our estimates of mechanical gradients of the tip-link complex. However, because 92% of our recordings were performed at P8 or later (Materials and methods), the tip-link complex ought to be nearly mature in our experiments, at least in outer hair cells (*Roth and Bruns, 1992*; *Waguespack et al., 2007*; *Beurg et al., 2018*). In inner hair cells, we cannot exclude that maturation of the hair-bundle morphology was still proceeding at the most apical cochlear positions explored in our study (*Peng et al., 2009*). Maturation sharpens the apex-to-base gradient of bundle height (*Roth and Bruns, 1992*); based on bundle morphology only, we would expect to underestimate stiffness gradients with immature inner hair cells.

Within the tip-link complex, transmembrane channel-like protein isoforms 1 and 2 (TMC1 and TMC2) are thought to be essential components of the transduction channels (*Fettiplace and Kim, 2014*; *Pan et al., 2018*). TMC2 is only transiently expressed after birth; TMC1 is expressed later than TMC2 but is fundamental to mechanoelectrical transduction of mature cochlear hair cells (*Kawashima et al., 2011*; *Kim and Fettiplace, 2013*). Stereociliary expression levels of TMC1, as well as their tonotopic gradients, were recently shown in mice to be nearly mature by P6, both in inner and outer hair cells, but TMC2 may still be present in apical inner hair cells until about P13 (*Beurg et al., 2018*). Because TMC2 confers larger $Ca^{2+}$ permeability to the transduction channels (*Kim and Fettiplace, 2013*), the $Ca^{2+}$ influx at rest in the inner hair cells at P8-P10 may have been larger than at more mature developmental ages, possibly lowering tip-link tension (*Figure 6*) and steepening its gradient.

## How stiffness gradients may contribute to the tonotopic map

We observed that the hair-bundle stiffness increased by 240% over two octaves (1–4 kHz) of characteristic frequencies for outer hair cells and by a similar amount but over 4 octaves (1–15 kHz) for inner hair cells (*Figure 2E*). Whether or not stiffness would continue increasing along the same gradient toward more basal locations of the cochlea is unknown. If it were the case, we would expect a base-to-apex stiffness ratio of ~40 for outer hair cells, which is comparable to the base-to-apex ratio of characteristic frequencies in the rat cochlea (range: 0.5–50 kHz; *Viberg and Canlon, 2004*), but only of ~6 for inner hair cells. The interplay between the stiffness and mass of a hair bundle could in principle help specify the preferred frequency of vibration of the hair cell through passive mechanical resonance with sound stimuli (*Frishkopf and DeRosier, 1983*; *Freeman and Weiss, 1990*; *Gummer et al., 1996*; *Holton and Hudspeth, 1983*; *Manley et al., 1988*). The resonance frequency

$\omega_C = \sqrt{k/m}$ of a spring-mass system is given by the square root of the system's stiffness $k$ divided by the mass $m$; it thus increases with stiffness, but relatively slowly. Assuming for simplicity that the bundle's mass remains nearly the same along the tonotopic axis (*Tilney and Tilney, 1988*), two orders of magnitude in frequency must be produced by a 10,000-fold increase in stiffness, corresponding to much steeper gradients than those reported here.

Alternatively, it has been proposed that the hair bundle could actively resonate with sound as the result of spontaneous oscillations (*Martin et al., 2001*; *Hudspeth, 2008*). Within this framework, the characteristic frequency is set by the frequency of the oscillator, which is expected to increase with the stiffness of the hair bundle (*Vilfan and Duke, 2003*; *Tinevez et al., 2007*; *Martin, 2008*; *Barral et al., 2018*). Notably, the relation may be steeper than that resulting from a passive spring-mass system, possibly approximating a linear dependence (*Hudspeth et al., 2010*). In this case, the stiffness gradient observed here (*Figure 2E*) for outer hair cells, but not for inner hair cells, could be steep enough to be a major determinant of the tonotopic map.

## Functional role of tension gradients

Tip-link tension is thought to control the open probability of the transduction channels, with higher tension promoting opening of the channels (*Hudspeth and Gillespie, 1994*). On this basis, a gradient of tip-link tension (*Figure 5C*) ought to result in a gradient of open probability. Yet, it has been shown in outer hair cells that the channels' open probability—the operating point of the transducer—remains remarkably uniform along the tonotopic axis, near a value of ½ (*Johnson et al., 2011*). To explain this observation, we note that the tension gradient for outer hair cells is associated with a gradient of single-channel conductance (*Beurg et al., 2006*; *Beurg et al., 2015*; *Beurg et al., 2018*). As a consequence, the magnitude of the $Ca^{2+}$ influx into transducing stereocilia is expected to increase with the characteristic frequency of the hair cell. Manipulations that affect the extracellular or the intracellular $Ca^{2+}$ concentration indicate that the transduction channels close at increased $Ca^{2+}$ concentrations (reviewed in *Fettiplace and Kim, 2014*), possibly because the channels are harder to open when the $Ca^{2+}$ concentration is higher near the channel's pore (*Cheung and Corey, 2006*). Thus, the gradient of tip-link tension reported here (*Figure 5C*) may compensate for the effects of the conductance gradient on the open probability: channels with higher conductance impart higher $Ca^{2+}$ influxes (closing the channels) but are also subjected to higher tension (opening the channels), perhaps maintaining an optimal operating point for the transducer at all cochlear locations.

Tension in the tip links is thought to be produced actively by pulling forces from molecular motors interacting with the actin core of the stereocilia at the upper insertion point of the tip link (*Gillespie and Müller, 2009*). The observed tension gradient in turn implies that, towards basal cochlear locations, there are more motors or that each motor exerts higher forces than near the apex. Notably, the tip links of inner-hair-cell bundles were found to bear less tension than those of outer-hair-cell bundles (*Figure 5B–C*). This property qualitatively makes sense, for the open probability of the transduction channels is thought to be smaller in inner hair cells than in outer hair cells (*Russell and Sellick, 1983*). There is also no, or only a weak, gradient of the single-channel conductance in inner hair cells (*Beurg et al., 2006*; *Beurg et al., 2018*), which parallels the relatively weak gradient of tip-link tension observed here.

## Tip-link tension may be high enough to alter tip-link conformation and affect gating-spring stiffness

The tip link is composed of the association of two cadherin-related proteins, cadherin-23 and protocadherin-15 (PCDH15) (*Kazmierczak et al., 2007*). Molecular dynamics simulations have suggested that a bend between extracellular cadherin (EC) repeats 9 and 10 of PCDH15 may confer some compliance to otherwise rigid tip links (*Araya-Secchi et al., 2016*). Tensions higher than ~10 pN are predicted to evoke complete unbending of EC9-10, resulting in significant stiffening of the tip link. Assuming that PCDH15 in the tip link forms a dimer (*Kazmierczak et al., 2007*; *Ge et al., 2018*) and that tip-link tension is equally shared by the two filaments, our estimates of tip-link tension (*Figure 5C*) are compatible with a contribution of the bending elasticity of EC9-10 to gating-spring stiffness at the apex of the rat cochlea, especially in inner hair cells. In outer hair cells, as one progresses from the very apex towards more basal cochlear locations, tension may quickly become too

high to allow a bent conformation in EC9-10. At the 4 kHz location, we estimated a resting tip-link tension of ~35 pN. Taking the measured unfolding forces of Ig domains in titin as a reference (*Rief et al., 1997*), tip-link tension might actually be high enough to evoke unfolding of EC domains, at least under resting conditions or at physiological loading rates. Recent evidence suggests that unfolding a various number of EC domains may contribute to a gradation of gating-spring stiffness (*Bartsch et al., 2018*; *Bartsch and Hudspeth, 2018*).

Notably, the estimated gradients of gating-spring tension (*Figure 5C*) were associated with gradients of gating-spring stiffness (*Figure 3D*): stiffer gating springs are subjected to more resting tension. Strain stiffening is a common phenomenon associated with the entropic elasticity of macromolecules, including the tip-link component PCDH15 (*Bartsch et al., 2018*), as well as with filamentous protein networks (*Bustamante et al., 1994*; *Rief et al., 1997*; *Kang et al., 2009*). A tension gradient may thus in part explain the existence of the observed gradient of gating-spring stiffness. Alternatively, the gating-spring stiffness could vary if the gating spring were composed of a variable number of compliant molecules operating in parallel and connected to a single tip link.

## Tip-link tension depends on calcium

Upon iontophoretic application of a $Ca^{2+}$ chelator (EDTA), before tip-link disruption, we observed that the hair bundle first moved in the negative direction and that this movement was associated with a concomitant opening of the transduction channels (*Figure 4*). Calcium acts as a permeant channel blocker of the transduction channels (*Fettiplace and Kim, 2014*). Lowering the extracellular $Ca^{2+}$ concentration is thus expected to increase the magnitude of the current flowing through open transduction channels but not to produce hair-bundle movements, at least as the result of block release only. A decrease of the extracellular $Ca^{2+}$ concentration also promotes opening of the transduction channels (*Hacohen et al., 1989*; *Johnson et al., 2011*). Within the framework of the gating-spring model of mechanoelectrical transduction, channel opening must reduce gating-spring extension and in turn tension, fostering *positive* movements of the hair bundle. Thus, the observed *negative* movements cannot result from internal forces associated with channel gating. Instead, our observations are readily explained if the evoked reduction of extracellular $Ca^{2+}$ concentration resulted in an increase of tip-link (and thus gating-spring) tension. If tip-link tension at rest is set by myosin molecular motors that pull on the tip links (*Hudspeth and Gillespie, 1994*), then the motor force must increase at decreased $Ca^{2+}$ concentrations.

Alternatively, a negative deflection would also be produced if lowering the extracellular $Ca^{2+}$ concentration evoked stiffening (*Martin et al., 2003*; *Beurg et al., 2008*) or shortening of the gating springs, increasing tension in the tip links. However, this mechanism would result in a negative steady-state offset of the bundle's resting position (*Figure 6E*) only if there were no or little myosin-based adaptation—usually called 'slow adaptation' (*Hudspeth and Gillespie, 1994*; *Peng et al., 2013*)—to relax the tension change and return the hair bundle near its initial resting position. Because we observed the mechanical correlate of slow adaptation in response to force steps (*Figure 2—figure supplement 5*), a $Ca^{2+}$-dependent regulation of tip-link tension via a change of gating-spring stiffness or length appears unlikely. This inference may be relevant to recent data implicating the stereociliary membrane as a key regulator of transduction-channel gating. Calcium ions have indeed been proposed to interact extracellularly with the local lipid environment of the transduction channels, promoting lower values of the channels' open probability (*Peng et al., 2016*). Our recordings (*Figure 4B* and *Figure 6D–E*) are consistent with this hypothesis but a change in membrane mechanics would not explain the observed steady-state increase in tip-link tension (*Figure 6E*) if slow adaptation happens in these cells (*Figure 2—figure supplement 5*).

Interestingly, depolarization of rat outer hair cells was previously shown to evoke positive movements of the hair bundle (*Kennedy et al., 2006*). Both depolarization and chelation of extracellular $Ca^{2+}$ are expected to reduce the intracellular $Ca^{2+}$ concentration in the vicinity of the transduction channel's pore. Yet, the directionality of active hair-bundle movements is opposite in the two studies, suggesting that the hair bundle can operate in two regimes (*Tinevez et al., 2007*). In the first regime (*Kennedy et al., 2006*), the response to $Ca^{2+}$ changes is dominated by gating forces (*Howard and Hudspeth, 1988*) so that the resting tension in the tip links is nearly the same before and after application of the stimulus. In the other regime (our study), $Ca^{2+}$-evoked changes of the resting tension in the tip links (*Figure 6*) dominate gating forces. In the chicken cochlea, depolarization of the hair cell was reported to evoke negative movements of the hair bundle (*Beurg et al.,*

*2013*), a directionality in agreement with that found here (*Figure 4A*). In addition, it has been shown in the bullfrog's sacculus (*Tinevez et al., 2007*) and the turtle's cochlea (*Ricci et al., 2002*) that the response of different hair cells to a given $Ca^{2+}$ change can be of either directionality and that the directionality of the response for a given hair cell can even be reversed by applying a position offset to the hair bundle. The two regimes of active hair-bundle motility can thus potentially coexist within the same hair cell, but only if gating forces are strong enough (*Tinevez et al., 2007*). We measured force-displacement relations that were remarkably linear (*Figure 2B and D*), showing no sign of gating compliance (*Howard and Hudspeth, 1988*). This observation confirms that gating forces were relatively weak under our experimental conditions, although others have shown that gating compliance can be measured with mammalian cochlear hair cells (*Russell et al., 1992*; *Kennedy et al., 2005*).

## Mechanical gradients reflect the division of labor between inner and outer hair cells

Stiffness (*Figure 2E*) and tension (*Figure 5B*) gradients were steeper for outer hair cells, which serve primarily as mechanical amplifiers of sound-evoked vibrations, than for inner hair cells, the true sensors of the inner ear (*Hudspeth, 2014*). Other properties, such as the height of the hair bundle (*Wright, 1984*; *Lim, 1986*; *Roth and Bruns, 1992*) or the conductance of the transduction channels (*Beurg et al., 2006*; *Beurg et al., 2018*), show a similar behavior. Thus, the division of labor between inner and outer hair cells may impart more stringent regulatory constrains to outer hair cells to tune their mechanoreceptive antenna according to the local characteristic frequency of the cochlear partition. However, the exact contribution of the hair bundle to frequency tuning remains unsure and, more generally, the mechanism that specifies the characteristic frequency remains a fundamental problem in auditory physiology. This may be in part because frequency selectivity cannot be ascribed to one element only, for instance the passive resonant property of the basilar membrane that was characterized in the pioneering work of von Békésy (*Von Békésy and Wever, 1960*). Various models of cochlear mechanics instead indicate that the characteristic frequency emerges from an active dynamic interplay between somatic electromotility of outer hair cells (*Ashmore, 2008*) and the micromechanical environment, including the basilar and tectorial membranes, as well as the hair bundle (*Nobili and Mammano, 1996*; *Hudspeth et al., 2010*; *O Maoiléidigh and Jülicher, 2010*; *Meaud and Grosh, 2011*; *Hudspeth, 2014*; *Reichenbach and Hudspeth, 2014*). Mechanical tuning of the inner constituents of the cochlear partition appears to happen at many scales: from the mesoscopic scale of the basilar and tectorial membranes, to the cellular scale of the hair bundle and hair-cell soma, down to the molecular scale of the hair cell's transduction apparatus. Our work demonstrates that tonotopy is associated, in addition to other factors, with stiffness and tension gradients of the tip-link complex.

## Materials and methods

### Experimental preparation

All experimental procedures were approved by the Ethics committee on animal experimentation of the Institut Curie; they complied with the European and French National Regulation for the Protection of Vertebrate Animals used for Experimental and other Scientific Purposes (Directive 2010/63; French Decree 2013–118). Experiments were performed on excised cochlear coils of Sprague Dawley rats (Janvier Labs) between postnatal day 7 and 10 (P7–P10), with 8% of the cells at P7, 75% at P8–P9 and 17% at P10. The dissection of the cochlea followed a published procedure (*Kennedy et al., 2003*). In short, we cracked open the bony shell covering the cochlear tissue, unwound the cochlear tube from the modiolus, removed the stria vascularis, and gently peeled the tectorial membrane. Apical or middle turns of the organ of Corti were positioned under strands of nylon fibers in the experimental chamber. We recorded from inner hair cells at four positions along the longitudinal axis of the cochlea (*Figure 1A*), corresponding to fractional distances of 5%, 10%, 20%, and 50% from the cochlear apex. According to the tonotopic map in this species (*Viberg and Canlon, 2004*), these cells were tuned at characteristic frequencies of 1, 2, 4, and 15 kHz, respectively. We also recorded from outer hair cells but only at the first three positions along the tonotopic axis. We have attempted to record from outer hair cells farther toward the cochlear base, in

particular at the 7 kHz location. However, our success rate was too low to get reliable estimates of tip-link tension and gating-spring stiffness, the primary goal of our work. Among possible reasons, basal outer hair cells might not withstand immersion in low-$Ca^{2+}$ (~20 μM) saline, a condition that we used to disrupt the tip links. In addition, the hair bundles of outer hair cells are smaller towards the base; monitoring their movements from direct imaging on photodiodes (see 'Microscopic apparatus' below) thus gets harder because the contrast of the image is lower.

The tissue was bathed in a standard saline containing 150 mM NaCl, 6 mM KCl, 1.5 mM $CaCl_2$, 2 mM Na-pyruvate, 8 mM glucose and 10 mM Na-HEPES. In some experiments, we used a low-$Ca^{2+}$ saline containing 150 mM NaCl, 6 mM KCl, 3.3 mM $CaCl_2$, 4 mM HEDTA, 2 mM Na-pyruvate, 8 mM glucose, and 10 mM Na-HEPES. As measured with a $Ca^{2+}$-sensitive electrode, this solution had a free $Ca^{2+}$ concentration of 22 μM, similar to that found in rat endolymph (*Bosher and Warren, 1978*). All solutions had a pH of 7.4 and an osmotic strength of 315 mOsm·$kg^{-1}$. Experiments were performed at a room temperature of 20–25°C.

## Microscopic apparatus

The preparation was viewed through a ×60 water-immersion objective of an upright microscope (BX51WI, Olympus). The tip of individual hair bundles was imaged at a magnification of ×1000 onto a displacement monitor that included a dual photodiode. Calibration was performed before each recording by measuring the output voltages of the monitor in response to a series of offset displacements of the photodiode. For hair-bundle movements that did not exceed ±150 nm in the sample plane, the displacement monitor was linear.

## Iontophoresis of a $Ca^{2+}$ chelator

We used iontophoresis to apply the calcium chelator EDTA in the vicinity of a hair bundle (*Figure 1B*) and disrupt its tip links (*Assad et al., 1991*; *Jaramillo and Hudspeth, 1993*; *Marquis and Hudspeth, 1997*). Coarse microelectrodes were fabricated from borosilicate capillaries with a pipette puller (P97, Sutter Instrument); their resistance was 1 MΩ when filled with 3 M KCl and immersed in the same solution. In experiments, the electrodes were filled with a solution containing 100 mM EDTA and 25 mM KCl. The electrode's tip was positioned at ~3 μm from the hair bundle. A holding current of +10 nA was continuously applied to counteract the diffusive release of EDTA from the electrode. The stimulus consisted of a −100-nA current step on top of the holding current, resulting in a net iontophoretic current of −90 nA. To facilitate tip-link disruption upon EDTA iontophoresis, the cochlear tissues were immersed in low-$Ca^{2+}$ saline (~20 μM $Ca^{2+}$).

## Mechanical stimulation and stiffness measurements

The hair bundles of inner and outer hair cells were mechanically stimulated using a fluid-jet device (*Kros et al., 1992*; *Géléoc et al., 1997*; *Johnson et al., 2011*). Pipettes were pulled from borosilicate glass (TW150-F, World Precision Instruments); their tip diameter was adjusted within a range of 5–10 μm. Fluid flow through a pipette was driven by a voltage command to a piezoelectric disk (Murata 7BB-27–4). Any steady-state flow coming in or out of the pipette was nulled by changing the hydrodynamic pressure inside the fluid-jet pipette; the hydrodynamic pressure was adjusted with a syringe connected to the body of the fluid-jet device. The fluid-jet pipette was positioned on the abneural side of the bundle along the hair bundle's axis of mirror symmetry (*Figure 1C*). Fluid coming out the pipette thus deflected the hair bundles towards the shortest stereociliary row, closing the ion channels that mediate mechanoelectrical transduction. This direction of bundle movement is defined as the negative direction in this paper; conversely, positive movements were directed towards the tallest row of stereocilia, fostering opening of the transduction channels. Mechanical stimuli were applied as 100-ms paired-pulse steps (*Figure 2* and *Figure 2—figure supplement 3*), or 60-Hz sinusoids (*Figure 4*) with the magnitude of driving voltages varying between 0 and 60 V.

For stiffness measurements, we measured hair-bundle movements evoked by 100-ms force steps (*Figure 2*; see the force-calibration procedure below). The bundle displacement was measured 5–10 ms after the onset of the step stimulus; the stiffness was given by the slope of the relation between the force (noted *F* in the following) and the displacement of the bundle's tip. These measurements were performed in standard saline.

## Applying and measuring forces with the fluid jet

We describe here how we calibrated the hydrodynamic drag force $F_D$ applied to the hair bundle by a fluid jet by using a flexible glass fiber of known stiffness as a reference. The method is based on a published procedure (*Géléoc et al., 1997*) that we refined to account for the non-uniform velocity field of the fluid (*Figure 2—figure supplement 1*). Using a generalized Stokes equation (*Leith, 1987*), the drag force can be written as $F_D = 6\pi\eta R_{HB} U$, in which $\eta$ is the viscosity of the surrounding fluid and $R_{HB}$ is the effective hydrodynamic radius of the bundle. The effective radius $R_{HB}$ was approximated by that of a prolate ellipsoid of short axis $h$ and long axis $W$, which correspond to the bundle's height and width, respectively. For a fluid flow perpendicular to the axis of rotational symmetry of the ellipsoid, this yields:

$$R_{HB} \cong 4h / \left\{ 3\left[ \phi/(\phi^2 - 1) + \left( (2\phi^2 - 3) \ln\left(\phi + \sqrt{\phi^2 - 1}\right)\right) / (\phi^2 - 1)^{3/2} \right] \right\}, \tag{1}$$

in which $\phi = W/h$ represents the aspect ratio of the ellipsoid (*Happel and Brenner, 2012*). We note that the hydrodynamic radius given by *Equation 1* is exact only for an ellipsoid immersed in an infinite volume of fluid, whereas the hair bundle instead stands erect at the apical surface of the hair cell; this expression is thus an approximation. *Figure 3—source data 1* and *Figure 3—figure supplement 2* recapitulate the values of parameters $h$ and $W$ that we used to model inner and outer hair-cell bundles along the tonotopic axis of the rat cochlea, as well as the resulting values of $R_{HB}$. The effective velocity $U \cong \int_{-W/2}^{W/2} v_X(x, y)\, dy / W$ was estimated by computing the mean of the velocity field $v_X(x, y)$ of the fluid over the width $W$ of the hair bundle. Here, $v_X(x, y) = \vec{v} \cdot \vec{e}_X$ is the projection of the fluid velocity $\vec{v}$ on the axis of mechanosensitivity (axis $X$) of the hair bundle; its value is estimated along the axis ($Y$) perpendicular to axis $X$ for a bundle positioned at a distance $x$ from the mouth of the fluid-jet pipette (*Figure 2—figure supplement 1A*). Using bead tracers, we found that the velocity profile $v_X(x, y)$ obeyed (*Schlichting, 1933*)

$$v_X(x, y) = V_{max}(x) / \left( 1 + (y/A(x))^2 \right)^2, \tag{2}$$

where $V_{max}(x)$ and $A(x)$ characterize, respectively, the maximal speed and the lateral extension of the velocity field at position $x$ (*Figure 2—figure supplement 1B–D*). By integrating the velocity profile, we obtain an expression for the force

$$F_D = 6\pi\eta R_{HB}\, \beta_{HB}\, V_{max}, \tag{3}$$

where $\beta_{HB} = \beta(w) = \frac{1}{2w}(w/(1 + w^2) + \tan^{-1} w)$ is a constant that depends on the normalized width of the hair bundle $w = W/(2A)$. Thus, calibrating the force $F_D$ is equivalent to calibrating the maximal fluid velocity $V_{max}$.

To estimate $V_{max}$, we measured the force $\overline{F}_D \cong 6\pi\eta R_F\, \overline{U}$ applied by the same jet on a calibrated glass fiber, whose longitudinal axis was oriented perpendicularly to that of the fluid-jet pipette. Given the diameter $D_F$ of the fiber, the effective hydrodynamic radius of a cylindrical fiber was calculated as $R_F = 2L/[3(\ln(L/D_F) + 0.84)]$ (*Tirado and de la Torre, 1979*). Because the conical fluid jet intersected the fiber over a length $L > W$, the effective fluid velocity $\overline{U} \cong \int_{-L/2}^{+L/2} v_X(x, y)\, dy / L = \beta_F\, V_{max}$ for the fiber was smaller than the effective velocity $U$ for the hair bundle, where $\beta_F = \beta(L/(2A)) < \beta_{HB}$. In practice, we used $L(x) = 2\, x\, \tan\alpha + D_{FJ}$, where $\alpha$ is the half-aperture of the conical fluid jet that was visualized using a dye (Coomassie Brilliant Blue; *Figure 2—figure supplement 2*) and $D_{FJ}$ is the diameter of the mouth of the fluid-jet pipette. We noticed that $L(x) \cong 2A(x)$ (*Figure 2—figure supplement 1*; *Figure 2—figure supplement 2*). We used this property to estimate $\beta_{HB} \cong \beta(W/L)$ and $\beta_F \cong \beta(1)$ without having to measure $A$ directly in every experiment.

In experiments, the projected horizontal distance between the tip of the fluid-jet pipette and the hair bundle or the fiber was fixed at $x \cong 8\ \mu m$ (mean ± SD: 7.8 ± 0.6 μm; range: 5.9–8.8 μm). Flexible fibers of diameters $D_F = 0.7$–1.5 μm and stiffness $k_F = 0.2$–2 mN/m were fabricated and calibrated as described before (*Bormuth et al., 2014*); their effective hydrodynamic radii varied within a range of $R_F = 2.5$–3.2 μm. A fluid jet of given magnitude elicited a force $\overline{F}_D = k_F \Delta X$, where $\Delta X$ is here the

measured deflection of the fiber. The relation between the force $\overline{F}_D$ applied to a fiber and the voltage command to the fluid-jet device was linear; its slope provided the calibration constant $C$ (*Figure 2—figure supplement 3*). When stimulating a hair bundle, a voltage command $V_C$ to the fluid-jet device thus elicited a force $F_D \cong GCV_C$, where $G = F_D/\overline{F}_D = (\beta_{HB}R_{HB})/(\beta_F R_F)$. We used $G = 1.4 \pm 0.1$ (mean $\pm$ SD; range: 1.27–1.65) for inner hair cells and $G = 1.3 \pm 0.1$ (mean $\pm$ SD; range: 1.12–1.47) for outer hair cells. Thus, we estimate that the force applied on the hair bundle was 30–40% higher than that measured on the calibration fiber using the same jet of fluid. In practice, we calculated $G$ in each experiment from the geometrical parameters of the fluid-jet pipette, the calibration fiber, and the hair bundle. Note that at a distance $y = L/2$ from the center of the fluid jet ($y = 0$), the fluid velocity is expected to be 25% of the maximal value (see *Equation 2* above). Thus, some of the moving fluid was not taken into account in our estimates of the force acting on the fiber, resulting in an underestimation. However, taking into account the velocity field up to $y = L$ would result in an increase of $G$ by only 5% while the fluid velocity at the edge of the fluid cone would be 4% of the maximal value .

All the forces reported in this work correspond to the effective force $F$ that one would have to apply at the bundle's tip to evoke the same bundle deflection $X$ as the hydrodynamic drag force $F_D$ actually exerted by the fluid jet and distributed over the height of the hair bundle. Correspondingly, the stiffness of the hair bundle was calculated as the slope of the force-displacement relation $F(X)$. In the following, we develop an approximate description of hair-bundle mechanics to relate the effective force $F$ to the drag force $F_D$. Modeling the hair bundle as a pivoting solid (*Kozlov et al., 2007*; *Kozlov et al., 2011*), we write the torque produced by the fluid jet as $\Gamma = \int_0^h f(z)\, z\, dz = \kappa_{HB}\, \theta$. Here, $h$ is the height of the hair bundle, $f$ is the drag force per unit (vertical) length experienced by the hair bundle at a vertical distance $z$ from the apical surface of the hair cell, $\kappa_{HB}$ is the rotational stiffness of the hair bundle, and $\theta \cong X/h$ is the pivoting angle corresponding to the deflection $X$ measured at the bundle's top. With these notations, the total drag force on the hair bundle reads $F_D = \int_0^h f(z)\, dz$. Assuming that the drag force per unit length $f \propto v(z)$ is proportional to the local fluid velocity $v(z)$ and that the condition of no-slip at the apical surface of the hair cell ($v(z = 0) = 0$) imposes a linear velocity profile $v(z) = v(h)\, z/h$, we take $f \propto z$. Injecting this expression into the integrals defining the torque $\Gamma$ and the total drag force $F_D$, we find $\Gamma = \frac{2}{3}F_D h$. As a result, we identify the effective force $F = \frac{2}{3}F_D$ that must be applied at the bundle's top to evoke the same torque as the fluid jet; the effective force $F$ is smaller than the total drag force $F_D$ exerted by the fluid jet, corresponding to a ratio $F/F_D = 2/3$.

To test our force-calibration procedure for the fluid jet, we performed a control experiment using hair-cell bundles from the sacculus of the frog (strain 'Rivan92' of *Rana ridibunda* [*Neveu, 2009*]). Details of the experimental preparation have been published elsewhere (*Tinevez et al., 2007*). In a similar preparation, the hair bundles have been shown to be very cohesive (*Kozlov et al., 2007*). As a result, the stereocilia are constrained to move as a unit in response to point-force application at the kinociliary bulb with a flexible fiber, allowing for well-controlled stiffness measurements (*Martin et al., 2000*; *Bormuth et al., 2014*). We compared stiffness estimates from stimulation with fluid jets and flexible fibers (*Figure 2—figure supplement 4*). From a sample of 13 hair bundles, we found a stiffness of $0.40 \pm 0.17$ mN/m (mean $\pm$ SD) using fluid-jet stimulation and of $0.42 \pm 0.16$ mN/m (mean $\pm$ SD) using fiber stimulation; the mean values of the stiffness estimates were not statistically different (paired-sample t-test). The good agreement between the two methods gave us confidence that the procedure that we used to calibrate the fluid jet is satisfactory.

## Electrophysiological recordings

We used the patch-clamp technique to measure mechano-electrical transduction currents. Borosilicate patch pipettes were filled with an intracellular solution containing 142 mM CsCl, 3.5 mM $MgCl_2$, 1 mM EGTA, 5 mM $Na_2$-ATP, 0.5 mM $Na_2$-GTP and 10 mM HEPES (pH = 7.3, 295 mOsmol/kg). When immersed in standard saline, these pipettes had a resistance of 1.5–3 M$\Omega$. A patch pipette was inserted in the organ of Corti through a pre-formed hole in the reticular lamina and approached parallel to the hair-cell rows toward the soma of a target hair cell. During the approach, standard saline was abundantly perfused to protect the $Ca^{2+}$-sensitive tip-links from EGTA. Hair cells were whole-cell voltage clamped at a holding potential of $-80$ mV; transduction currents were low-pass filtered at 1–10 kHz (Axopatch 200B; Axon Instruments). No correction was made for the liquid-

junction potential. The series resistance was always below 10 MΩ and was compensated up to 70%. To disrupt the tip links with EDTA iontophoresis, the solution bathing the cells was changed to low-$Ca^{2+}$ saline after the cell was patched; the solution change was performed either with a perfusion or with a Picospritzer (Picospritzer III, Parker).

## Scanning electron microscopy

Cochleae from P8 rats were processed with osmium tetroxide/thiocarbohydrazide, as previously described (*Furness et al., 2008*). Samples were analyzed by field emission scanning electron microscopy operated at 5 kV (Jeol JSM6700F). The number of stereocilia in inner and outer hair-cell bundles was estimated from electron micrographs at each of the cochlear locations where we performed mechanical and electrophysiological measurements (*Figure 3—figure supplement 2*; *Figure 3—source data 1*).

## Estimating the number of intact tip links in a hair bundle

We performed patch-clamp recordings of the transduction current $I_{MAX}$ elicited at saturation by large hair-bundle deflections (*Figure 3—figure supplement 3*). In inner hair cells, the number of intact tip links $N_{TL} = I_{MAX}/I_1$ was calculated by dividing the saturated current $I_{MAX}$ for the whole hair bundle by the published estimate $I_1 = 35.4$ pA for the transduction current flowing through the tip of a single transducing stereocilium (*Beurg et al., 2009*); electron microscopy has indeed shown that there is precisely one tip link per stereocilium in an intact hair bundle (*Pickles et al., 1984*; *Kachar et al., 2000*). We used the same value of $I_1$ at all cochlear locations (*Beurg et al., 2006*; *Beurg et al., 2018*). Given the magnitude $i = 15$ pA of the current flowing through a single transduction channel, there was on average $I_1/i = 2.36$ transduction channels per transducing stereocilium (*Beurg et al., 2006*). In outer hair cells, there is no direct estimate of $I_1$. However, the unitary current $i$ was shown to increase (*Beurg et al., 2006*; *Beurg et al., 2015*; *Beurg et al., 2018*) from 8.3 pA to 12.1 pA when the hair cell's characteristic frequency increases from 4 kHz to 14 kHz (*Beurg et al., 2006*). All these currents were measured under the same experimental conditions as ours, in particular using a −80 mV holding potential and with the hair cells immersed in a standard saline containing 1.5 mM $Ca^{2+}$. Assuming a linear relation between the unitary current and the position of the hair cell along the tonotopic axis of the cochlea (*Beurg et al., 2015*; *Beurg et al., 2018*), we inferred the unitary currents at other cochlear locations. We then assumed that the average number of transduction channels per tip link was 2.36, as estimated in inner hair cells (*Beurg et al., 2009*). The number of intact tip links was then calculated as $I_{MAX}/(2.36\,i)$. We performed this measurement for 10 hair cells at each cochlear location, both for inner and outer hair cells, to calculate the average number of intact tip links in any given hair cell. In these experiments, the hair cells were immersed in standard saline.

Recent measurements in the mouse cochlea have revealed that unitary currents may represent an ensemble average over multiple conductance states, raising the possibility that these currents are produced by a few (up to 5) identical transduction channels that gate cooperatively (*Beurg et al., 2018*). This finding does not affect our estimates: the current that flows through a single stereocilium stays the same, whether or not it results from cooperative gating of multiple channels or from gating of an effective channel endowed with the same conductance as the total conductance of the group.

## Signal generation and acquisition

All signals were generated and acquired under the control of a computer running a user interface programmed with LabVIEW software (version 2011, National Instruments). Command signals were produced by a 16-bit interface card at a sampling rate of 25 kHz (PCI-6733, National Instruments). A second 16-bit interface card (PCI-6250, National Instruments) conducted signal acquisition. Sampling rates for signal generation and acquisition varied within the range 2.5–25 kHz. All signals were conditioned with an eight-pole Bessel antialiasing filter adjusted to a low-pass half-power frequency at half the sampling rate of signal acquisition.

## Statistical significance

Unless otherwise noted, all results are quoted as mean ± standard error of the mean (*n*) with a number *n* of cells of at least 10 per group. G-Power analysis ensured that this number was sufficient to

achieve a signal-to-noise ratio of 1–1.5, with 80% power at a 5% significance level. We performed a one-way ANOVA to assay statistical significance of the measured mean-value variation of a given property, for example the hair-bundle stiffness, between the different cochlear locations for inner (IHC) or outer (OHC) hair cells. We also used two-tailed unpaired Student's $t$-tests with Welch's correction when comparing mean values between two groups of a given hair-cell type (IHC or OHC) with different characteristic frequencies or between the two cell types (IHC/OHC) with a given characteristic frequency. Stars correspond to p-values with *p < 0.05, **p < 0.01, and ***p < 0.001, whereas 'n.s.' (p > 0.05) indicates non-significant differences. To determine whether variables estimated from the product of $M$ independent variables $X_i$ ($i = 1 .. M$) had means that were statistically different, we first calculated the standard error of the mean $\sigma_P$ of the product and the effective number of degrees of freedom $\nu_{\text{eff}}$ of the product. Defining $\overline{X_i}$ the mean value, $s_i$ the standard deviation, and $\sigma_i = s_i/\sqrt{n_i}$ the standard error of the mean of variable $X_i$ over $n_i$ measurements, the standard error of the mean for the product was calculated as $\sigma_P = \prod \overline{X_i} \sqrt{\sum \left(\frac{\sigma_i}{\overline{X_i}}\right)^2}$ and the effective number of degrees of freedom associated with the product was calculated using the Welch-Satterthwaite approximation as $\nu_{\text{eff}} = \left[\sigma_P/\prod \overline{X_i}\right]^4 / \sum \frac{1}{n_i-1}\left(\frac{\sigma_i}{\overline{X_i}}\right)^4$. Finally, we characterized tonotopic gradients by performing weighted linear regressions, in which the weight applied to each data point was given by the inverse of the squared standard error of the mean. We then applied a $t$-test on the resulting coefficients to determine whether the observed difference between the gradients measured with inner and outer hair cells was statistically significant. The results of all statistical analyses are listed in tables which are provided as source data associated to the main figures.

## Acknowledgements

We thank Benoît Lemaire and Rémy Fert from the machine shop of the Curie Institute, Prof. Walter Marcotti for help in the design of our fluid-jet device, Jérémie Barral for a critical reading of the manuscript and Christine Petit for fruitful discussions. This research was supported by the French National Agency for Research (ANR-11-BSV5 0011 and ANR-16-CE13-0015) and by the Labex Celtisphybio ANR-10-LABX-0038. MT is an alumnus of the Frontiers in Life Science PhD program of Université Paris Diderot and thanks the Fondation Agir pour l'Audition for a doctoral fellowship. AC was supported by a PhD fellowship from the European Union Horizon 2020 research and innovation program under the Marie Skłodowska-Curie grant agreement No 66600.

## Additional information

### Funding

| Funder | Grant reference number | Author |
| --- | --- | --- |
| French National Research Agency | ANR-11-BSV5-011 | Pascal Martin |
| French National Research Agency | ANR-16-CE13-0015 | Pascal Martin |
| European Union Horizon 2020 | Marie Sklodowska-Curie grant No 66600 | Pascal Martin |
| Labex Celltisphybio part of the Idex PSL | ANR-10-LABX-0038 | Atitheb Chaiyasitdhi |

The funders had no role in study design, data collection and interpretation, or the decision to submit the work for publication.

### Author contributions

Mélanie Tobin, Conceptualization, Data curation, Formal analysis, Validation, Investigation, Visualization, Methodology, Writing—original draft, Writing—review and editing; Atitheb Chaiyasitdhi, Data curation, Validation, Investigation, Visualization, Methodology, Writing—review and editing, Stiffness measurements in frog; Vincent Michel, Data curation, Validation, Investigation, Visualization,

Methodology, Writing—review and editing, SEM imaging; Nicolas Michalski, Conceptualization, Supervision, Validation, Investigation, Visualization, Methodology, Writing—review and editing; Pascal Martin, Conceptualization, Resources, Data curation, Formal analysis, Supervision, Funding acquisition, Validation, Visualization, Methodology, Writing—original draft, Project administration, Writing—review and editing

### Author ORCIDs
Mélanie Tobin (iD) http://orcid.org/0000-0003-1669-1574
Atitheb Chaiyasitdhi (iD) https://orcid.org/0000-0001-8293-5683
Nicolas Michalski (iD) https://orcid.org/0000-0002-1287-2709
Pascal Martin (iD) https://orcid.org/0000-0001-6860-4677

### Ethics

Animal experimentation: All experimental procedures were approved by the Ethics committee on animal experimentation of the Institut Curie; they complied with the European and French National Regulation for the Protection of Vertebrate Animals used for Experimental and other Scientific Purposes (Directive 2010/63; French Decree 2013-118).

### Decision letter and Author response

Decision letter https://doi.org/10.7554/eLife.43473.029
Author response https://doi.org/10.7554/eLife.43473.030

## Additional files

### Supplementary files
• Transparent reporting form
DOI: https://doi.org/10.7554/eLife.43473.027

### Data availability
All data generated or analysed during this study are included in the manuscript and supporting files. Source data files have been provided for Figures 2, 3, 5 and 6.

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
