## [Decision Letter]

Thank you for sending your article entitled "Tonotopy of the mammalian cochlea is associated with stiffness and tension gradients of the hair cell's tip-link complex" for peer review at *eLife*. Your article is being evaluated by Andrew King as the Senior Editor, a Reviewing Editor, and three reviewers.

To convincingly demonstrate that the stiffness and tension in tip-link complex contributes to the tonotopy of the cochlea, reviewer #1 suggested that testing at 7 kHz location is warranted. Although the authors have stated that testing outer hair cells farther towards the basal cochlea was technically difficult, it is not clear whether testing CF around 7kHz has been attempted. Without an additional CF higher than 4 kHz, it may be difficult to convince reviewer #1. Please respond within the next two weeks with an action plan and timetable for the completion of the additional work requested by all the reviewers. We plan to share your responses with the reviewers and then issue a binding recommendation on whether to invite a revision.

*Reviewer #1:*

This is a meticulous study of the mechanical properties of the hair bundles of rat cochlear hair cells. During transduction, hair bundle displacement opens mechanotransducer (MT) channels via increased tension in tip links connecting adjacent stereocilia. It is thought that each tip link is connected, directly or indirectly, to an MT channel. While the paper contains much quantitative material, the most important conclusion is that the mechanical stiffness of the tip-link complex increases towards the high-frequency end of the cochlea. This is surprising given that each tip link is composed a dimer of protocadherin15 connected to a dimer of cadherin 23, the stiffness of which should be independent of location. The paper also includes measurements of the MT current and bundle stiffness versus cochlear location, and the effects of calcium, though none of this is really new. A significant weakness of the paper is that only a small length, the apical 20 percent, of the cochlea is sampled to derive a tonotopic gradient for outer hair cells, the conclusion being extrapolated to the remaining 80 percent. It is unclear why more basal positions were not tested when earlier work (Beurg et al., 2006) had shown this to be feasible. To provide context, the authors study only 1 to 4 kHz, where the rat upper frequency limit is at least 50 kHz. I was also concerned with the tonotopic map derived from Viberg and Canlon, (2004), when the direct measurements were present in Muller, (1991). Muller did not measure CFs less than 4 kHz for the young (P13) rats, and using his values, I would assess CFs, of 2, 4 and 7 kHz for the three OHC locations. Another weakness was that the stiffness values do not agree with previous measurements (Geleoc et al., 1997 or Beurg et al., 2008). For example, this paper reports an OHC bundle stiffness of 12 mN/m at CF= 4kHz, whereas Beurg et al., (2008) reported 3.1 mN/m and Geleoc et al., 4.5 mN/m. The latter work was done with the same technique as used here. The rapid increase between 2 kHz and 4 kHz (Figure 3A) is suspicious. In view of these concerns, I feel the paper would be better placed in a less high-profile journal such as Biophysical Journal or JARO.

1) Impact statement: “The mechanical properties of the tip-link complex contribute to frequency selectivity of auditory hair cells”. See also subsection: “Tip-link complex plays a role in setting the CF of the hair cell.” To my knowledge there is no evidence to support these contentions. Moreover, the present consensus (see e.g., work by Grosh and colleagues) is that prestin-based somatic contractility is sufficient to account for cochlear amplification.

2) Discussion section. “Computing the relative difference in stiffness ratio between the two extreme cochlear locations”: are 1 and 4 kHz regarded as extreme locations?

3) Discussion section. The section relating stiffness to MT channel number is an intriguing suggestion, but the correlation is poor and not enough of the cochlea is examined. Is it possible there is an error either in determining location or allocating CFs?

4) Discussion section. Whether or not the channel gating force is manifest in the bundle's force-displacement plot depends mainly on the bundle's passive stiffness. For example, there is a strong non-linearity for the frog saccular hair bundle, where the passive stiffness is less than 0.5 nmN/m (Cheung and Corey, 2005). For mammalian hair bundles, less stiff vestibular bundles show more non-linearity than more stiff auditory bundles (Geleoc et al., 1997). The present results with stiffness of 12 mN/m would not be expected to show any non-linearity. Please add this.

*Reviewer #2:*

It has been recognized for more than a century that the frequency sensitivity of the cochlea varies along the organ's length, a property that is essential for our ability to recognize various sounds. The bases of this so-called tonotopic mapping have nevertheless remained uncertain. The basilar membrane itself displays a gradient in its mechanical properties, but that is far too shallow to account for the human hearing range of 20 Hz to 20 kHz. Individual hair bundles likewise vary systematically in their heights and numbers of stereocilia; again, these gradients cannot explain the observed gamut of frequencies. The conductances of individual transduction channels of hair cells-especially outer hair cells-show a tonotopic gradient, but that phenomenon seems to have no direct bearing on frequency selectivity.

The present submission documents a novel and somewhat unexpected contribution to frequency tuning: a tonotopic gradient in the stiffness of hair bundles, in the stiffness of individual tip links, and in the tension that the links bear at rest. The work therefore constitutes an important contribution on a key issue in auditory physiology. The experimentation is of high quality and excellent statistical significance; the results are described clearly and succinctly.

There is a problematic issue in subsection “The hair-bundle stiffness increases along the tonotopic axis”. It is surprising a first glance that the authors mention neither adaptation and the consequential changes in hair-bundle stiffness, nor gating compliance owing to the opening of transduction channels. This is worrisome inasmuch as the lack of these signatures might imply unhealthy cells. Only in subsection “Mechanical stimulation and stiffness measurements” do the authors report the time-dependent changes characteristic of adaptation, which they later demonstrate in Figure 2—figure supplement 4. Moreover, they subsequently mention the period of 5-10 ms after the onset of stimulation at which measurements were made, an interval likely to miss gating compliance owing to the swift adaptation of mammalian hair cells. Note in that context the transient at the outset of the single record with a high temporal resolution (Figure 2—figure supplement 3). Finally, in subsection “Tip-link tension depends on calcium” they state that "gating forces were relatively weak under our experimental conditions," when really their measurement conditions probably precluded the detection of these forces. It would be useful to state at the outset that the principal measurements reflect "steady-state" stiffnesses or the like, and perhaps to note that adaptation was probably observed but was not relevant, and that gating compliance might have been missed for technical reasons.

*Reviewer #3:*

This is an interesting manuscript that investigates tonotopic differences in hair bundle mechanical properties between inner and outer hair cells. The data are interesting, novel and rigorous. The work provides an important contribution to our knowledge base. Data show that OHCs are different from IHCs and that there is a tonotopic gradient for OHC properties. A novel finding of a BAPTA sensitive force also providing a tonotopic variation for OHCs is described and discussed. Overall a very good piece of work that provides important new information about mammalian cochlear hair bundles.

1) It is important to be clear that the changes observed with BAPTA treatment are not necessarily limited only to tip-link proteins but anything in series with these proteins. The tonotopic changes described may be a reflection of intrinsic tip link properties but they also can be a reflection of changes described for the TMC proteins, upper insertion point proteins and even the lipid bilayer. The authors do allude to this idea in several places (using the term tip link complex), but it comes across a bit confused and it is unclear when they mean only the tip-link proteins and when they include accessory structures. The discussion of potential intrinsic tip- link protein effects is reasonable, it simply needs to be expanded to be inclusive as there is no clear data one way or the other.

2) Does the tonotopic change in stereocilia height alter the force sensed from the fluid jet because of the apical surface either reflecting flow or adding a resistance to flow? That is, how sure are the authors that the stimulus to the hair bundle is constant between tonotopic positions?

3) Is it possible that the resting tip-link tension variations are a function of tonotopic membrane potential differences operating in a feedback to control driving force and perhaps calcium permeation? Similarly, is it possible that previously described tonotopic differences in calcium buffering might underlie the measured changes in resting tension?

4) Is there concern that the ages used overlap in timing with changes in TMC proteins switching? As development occurs tonotopically, might some of the observations here be due to different developmental states?

5) In subsection “Tip-link tension depends on calcium” the authors appear to be making a statement about adaptation and calcium effects. If they want to go this way, then they should be as quantitative here as they are in the rest of the manuscript. As the data presented here is consistent with that from the Peng papers where calcium dependent and independent effects are described, the calcium dependent effects are simply argued to be separate from adaptation, just as the upper tip link insertion motors are argued to be a tensioning mechanism independent of adaptation. Seems to me that the data presented in this manuscript does not actually directly address the topic of calcium and adaptation. The authors should either make a statement that includes a conclusion, present data on the topic, or eliminate the paragraph.

6) The last sentence of the manuscript implies that the tonotopic variations in the hair bundle mechanics support an argument that the hair bundle is part of the active process. It seems to me that the data in the manuscript argue that OHCs, which need to work cycle by cycle are mechanically different tonotopically and also from IHCs which do not need to operate cycle by cycle. The data supports an argument for the hair bundle providing some frequency selectivity, perhaps a tuning mechanism, which may or may not be active. It might be more valuable for the authors to better delineate what they are referring to when they say active process, do they mean amplification, do they mean tuning, do they mean both? Seems like a very loaded sentence.

[Editors' note: further revisions were requested prior to acceptance, as described below.]

Thank you for resubmitting your work entitled "Stiffness and tension gradients of the hair cell's tip-link complex in the mammalian cochlea" for further consideration at *eLife*. Your revised article has been favorably evaluated by Andrew King (Senior Editor) and a Reviewing Editor and will be accepted for publication.

However, two of the reviewers remain concerned about the claims relating to "tonotopy" in the manuscript. We accept the reasons given for not being able to obtain responses at the 7 kHz location, but this does mean that only 20% of the length of the cochlea was assessed. We therefore think it would be prudent to stress this in the paper to avoid giving a misleading impression about the range of frequency locations that were sampled.

---

## [Author Response]

[Editors' note: the authors’ plan for revisions was approved and the authors made a formal revised submission.]

Reviewer #1:This is a meticulous study of the mechanical properties of the hair bundles of rat cochlear hair cells. During transduction, hair bundle displacement opens mechanotransducer (MT) channels via increased tension in tip links connecting adjacent stereocilia. It is thought that each tip link is connected, directly or indirectly, to an MT channel.While the paper contains much quantitative material, the most important conclusion is that the mechanical stiffness of the tip-link complex increases towards the high-frequency end of the cochlea. This is surprising given that each tip link is composed a dimer of protocadherin15 connected to a dimer of cadherin 23, the stiffness of which should be independent of location.

Finding that the stiffness associated with the tip-link complex –the gating-spring stiffness− increases with the characteristic frequency of the hair cell indeed constitutes a major result of our work. The reviewer seems to imply that the stiffness of the tip-link complex is determined by that of the tip link, i.e. that the tip link embodies the gating spring. As also suggested by reviewer 3 (point 1), we have clarified in the Introduction that the stiffness of the ‘tip-link complex’ cannot be reduced to that of the tip link itself, for the complex includes other proteins in series with it that may be more compliant.

“Sound evokes hair-bundle deflections, which modulate the extension of elastic elements −the gating springs− connected to mechanosensitive ion channels. […] In the following, the molecular assembly comprising the tip link, the transduction channels, as well as the molecules to which they are mechanically connected will be called the ‘tip-link complex’.”

If the gating spring lies in series with the tip links, our results would imply a progressive change in the chemical composition of the gating spring or in the number of parallel compliant elements attached to a tip link. Notably, the tip link is still discussed as a possible component of the gating spring (Hudspeth, 2014; Araya-Secchi et al., 2016; Bartsch et al., 2018; Bartsch and Hudspeth, 2018), in particular because its detailed structure at physiological Ca^2+^ concentrations (~20 µM) and resting tension is unknown and also because there hadn’t been direct estimates of its stiffness through force measurements until very recently ((Bartsch et al., 2018); see below). An important finding of our work is that the estimated gradients of gating-spring stiffness are associated with gradients of tip-link tension: stiffer gating springs are subjected to more resting tension. As pointed out in subsection “Tip-link tension depends on calcium”, there could be a causal relation between the two observations if the cadherin-related molecules that constitute the tip link displayed strain stiffening, a phenomenon associated with entropic elasticity. Remarkably, this hypothesis finds strong experimental support in a preprint recently posted by the group of Jim Hudspeth in BioArXiv (https://www.biorxiv.org/search/10.1101%252F503029). One reads in the abstract of this preprint: “[…], we show that an individual monomer of PCDH15 acts as an entropic spring […]. The tip link’s entropic nature then allows for stiffness control through modulation of its tension.” Thus, finding a gradient of gating-spring stiffness may not be that surprising, even in the case where the tip link contributes to the gating spring: a uniform molecular composition of the tip link (PCDH15-CDH23) along the tonotopic axis does not necessarily imply that the stiffness of the tip link is everywhere the same.

The paper also includes measurements of the MT current and bundle stiffness versus cochlear location, and the effects of calcium, though none of this is really new.

As explained in the Material and methods section, the saturating magnitude of MT currents are used here to estimate the number of intact tip links (noted *N_TL_)* that contribute to stiffness and tension of a hair bundle. These measurements are not a result per se but are nonetheless important in the procedure to estimate single gating-spring stiffness and tension, as well as their gradients, leading to the major findings of our work. Measurements of MT current are also essential to discuss the origin of the negative hair-bundle movements that are observed at the onset of an EDTA iontophoritic step (Figure 4B) (see subsection “Tip-link tension depends on calcium”).

In their pioneering work from the early 1980s (Strelioff and Flock, 1984), Strelioff and Flock first reported the existence of tonotopic gradients of hair-bundle stiffness in a mammalian cochlea, using the guinea pig as a model system. We confirm here the existence of tonotopic stiffness gradients of the whole hair bundle in the rat cochlea but also push much further: (i) by parsing out the relative contributions of the gating springs and the stereociliary pivots to the whole hair-bundle stiffness and (ii) by estimating stiffness and mechanical tension both at the level of the whole hair bundle and of a single gating spring. In addition, our method of hair-bundle stimulation (fluid jet moving the whole bundle) may be more controlled than fiber stimulation in these early measurements (the fiber pushed on a few stereocilia only), likely resulting in more reliable stiffness estimates.

Finally, we are not aware of any other study showing that (relatively) rapid changes in the *extracellular* Ca^2+^ concentration, here via iontophoresis of a Ca^2+^ chelator, can evoke active movements of cochlear hair bundles of the mammalian cochlea. Similar observations have been reported with hair cells from the bullfrog’s sacculus (e.g. in Tinevez et al., (2007)), but not with cochlear hair cells from a mammalian cochlea. This observation is thus new.

A significant weakness of the paper is that only a small length, the apical 20 percent, of the cochlea is sampled to derive a tonotopic gradient for outer hair cells, the conclusion being extrapolated to the remaining 80 percent. It is unclear why more basal positions were not tested when earlier work (Beurg et al., 2006) had shown this to be feasible. To provide context, the authors study only 1 to 4 kHz, where the rat upper frequency limit is at least 50 kHz.

We don’t see any stiffness measurement in Beurg et al., 2006; the reviewer probably refers to Beurg et al., (2008).

With OHCs, we are perfectly aware that adding data at a more basal position than the 4-kHz location, as we managed to do with IHCs, would be ideal. We recall that the main goal of our work is to provide, for the first time, estimates of tip-link tension and gating-spring stiffness in hair-cell bundles from a mammalian cochlea and to characterize their gradients along the tonotopic axis. Both tension and stiffness estimates rely on our ability to disrupt the tip links and, respectively, to measure an offset of hair-bundle position and a change in deflection amplitude upon fluid-jet stimulation.

We have actually attempted to disrupt the tip links of OHC bundles at the 7 kHz location. Accordingly, we have measured the response of ~50 OHCs from 9 ears (9 different animals) to EDTA iontophoresis. However, only two cells in one ear showed a noticeable hair-bundle movement, which qualitatively resembled that shown in Figure 4A but displayed a very small amplitude. This success rate was too low to go on with experiments at this location. By comparison, at more apical locations (1-, 2- and 4-kHz locations), EDTA iontophoresis evoked robust hair-bundle movements in 45% or more of the cochleae tested. The reason for our lack of success at the 7-kHz location is unclear. In our one-compartment preparation of the cochlea, OHCs at the 7-kHz location and further towards the cochlear base might not withstand immersion in low-Ca^2+^ (20 µM) saline. In addition, we noticed that voltage signals produced by our position monitor (photodiodes) appeared very noisy compared to signals measured at more apical positions. The calibration constant of our photodiode system was only 4.6 ± 1.1 mV/nm (n = 11) at the 7-kHz location, which was 60% lower than the value 11.0 ± 5 mV/nm (n = 12) measured at the 4 kHz. The hair bundles of OHCs are smaller towards the base; monitoring their movements from direct imaging on photodiodes thus gets harder because the contrast of the image is lower.

Because EDTA iontophoresis did not evoke robust hair-bundle movements at the 7 kHz position, we were unable to probe tip-link tension and gating-spring stiffness at this location and thus to reach our primary goal. For this reason, we did not perform stiffness measurements of the whole hair bundle at the 7-kHz location. This does not mean that the hair-bundle stiffness cannot be measured there. As pointed out by reviewer 1, Fettiplace’s group has published (though not in Beurg et al., (2006), but in Beurg et al., (2008)) stiffness measurements in 6 OHCs at a more basal location of the rat cochlea than those reported here, corresponding to a characteristic frequency of 14-kHz or equivalently to a 50% fractional distance along the cochlear axis. Although Beurg et al’s work is about probing the effects of Ca^2+^ on hair-bundle mechanics, we note that these measurements were performed at 1.5-mM Ca^2+^, not at endolymphatic concentrations (20 µM); this paper reports stiffness at low Ca^2+^ concentrations (20 µM) at the 4-kHz location only. It is tempting to speculate that, as we experienced, probing hair-bundle mechanics at more basal locations is very difficult, at least when the hair bundles are exposed to a low-Ca^2+^ saline. In our work, all tension measurements were performed at 20-µM Ca^2+^, which is near the natural Ca^2+^ concentration in endolymph (our results (Figure 6) show that tip-link tension depends on Ca^2+^). Note that EDTA iontophoresis as a means to disrupt tip links works well at 20-µM Ca^2+^, but not at 1.5-mM Ca^2+^ because the EDTA concentration cannot be raised high enough with iontophoresis to lower the Ca^2+^ concentration down to sub-μM levels.

To clarify why we do not include data beyond the 4-kHz position towards the cochlear base with outer hair cells, we have added the following paragraph in the subsection “Experimental preparation”:

“We have attempted to record from outer hair cells farther towards the cochlear base, in particular at the 7 kHz location. However, our success rate was too low to get reliable estimates of tip-link tension and gating-spring stiffness, the primary goal of our work. Among possible reasons, basal outer hair cells might not withstand immersion in low-Ca^2+^ (~20 µM) saline, a condition that we used to disrupt the tip links. In addition, the hair bundles of outer hair cells are smaller towards the base; monitoring their movements from direct imaging on photodiodes (see ‘Microscopic apparatus’ below) thus gets harder because the contrast of the image is lower.”

I was also concerned with the tonotopic map derived from Viberg and Canlon, (2004), when the direct measurements were present in Muller, (1991). Muller did not measure CFs less than 4 kHz for the young (P13) rats, and using his values, I would assess CFs, of 2, 4 and 7 kHz for the three OHC locations.

We don’t see where in this paper (Müller, (1991)) the author refers to a difference in the tonotopic map of young rats with respect to adult rats. We have indeed considered the tonotopic map measured in adult rats (Figure 2 in this paper; tonotopic map: x(f) = 102.048 exp(-0.04357 CF) – 4.632, as reproduced in Viberg and Canlon, (2004)). Note that others have used the same tonotopic map as we did (e.g. Beurg et al., (2008)).

Another weakness was that the stiffness values do not agree with previous measurements (Geleoc et al., 1997 or Beurg et al., 2008). For example, this paper reports an OHC bundle stiffness of 12 mN/m at CF= 4kHz, whereas Beurg et al., (2008) reported 3.1 mN/m and Geleoc et al., 4.5 mN/m. The latter work was done with the same technique as used here. The rapid increase between 2 kHz and 4 kHz (Figure 3A) is suspicious.In view of these concerns, I feel the paper would be better placed in a less high-profile journal such as Biophysical Journal or JARO.

Following the reviewer’s criticism, we have carefully reevaluated our calibration procedure of the fluid jet. This led to the identification of a systematic error corresponding to the overestimation of the forces applied by the fluid jet (and thus stiffness) by a factor ~3/2; all the data, statistical tests and figures were updated to account for this error. As detailed below, we have validated our calibration procedure by performing a control experiment in which we compared stiffness measurements with the fluid jet and with a flexible fiber using hair cells from the frog sacculus. We thank the reviewer for helping us find this error and thus improve the reliability of our stiffness estimates. Note that a systematic error in force measurements does not affect the conclusions of our paper, e.g. the existence of gradients in gating-spring stiffness and tension. After correction, the hair-bundle stiffness in outer hair cells at the 4-kHz position is found to be 8.6 mN/m. This value is still higher than those reported by Geleoc et al., 1997 and Beurg et al., (2008) but we explain below why the published values may represent underestimates of the actual stiffness.

We have added the following two paragraphs in the subsection Applying and measuring forces with the fluid jet”:

“All the forces reported in this work correspond to the effective force *F* that one would have to apply at the bundle’s tip to evoke the same bundle deflection *X* as the hydrodynamic drag force FD actually exerted by the fluid jet and distributed over the height of the hair bundle. […] From a sample of 13 hair bundles, we found a stiffness of 0.40 ± 0.17 mN/m (mean ± SD) using fluid-jet stimulation and of 0.42 ± 0.16 mN/m (mean ± SD) using fiber stimulation; the mean values of the stiffness estimates were not statistically different (paired-sample t-test; ***p<0.001). The good agreement between the two methods gave us confidence that the procedure that we used to calibrate the fluid jet is satisfactory.”

We now discuss why previous stiffness estimates from the literature may be different than ours. In Geleoc et al. 1997, the authors probed hair-bundle mechanics in neonatal OHCs (P1-2) from the mid-apical coil of the mouse cochlea after these cells had been maintained in culture for 1-4 days. We worked instead on acute preparations of the rat cochlea at P7-10. Geleoc’s preparation is thus quite different from ours and whether stiffness estimates can be directly compared between the two preparations is unclear. In addition, these authors considered that the velocity of the fluid coming out of their fluid-jet pipette is uniform along an axis perpendicular to the longitudinal axis of the pipette and that the jet is cylindrical. In contrast, our detailed characterization of the velocity field produced by a fluid jet demonstrates that the jet is conical and that the fluid-velocity profile is peaked (Figure 2—figure supplement 1 and Figure 2—figure supplement 2). Ignoring these features results in an underestimate of the drag force that is applied on a hair bundle, and in turn of the hair-bundle stiffness. This may explain, at least in part, why our stiffness estimates are higher than in Geleoc et al., 1997. In addition, Geleoc et al., estimated that the effective point force that one would have to apply at the bundle’s top to get the same deflection as that evoked by the fluid jet was half the total drag force applied by the fluid jet. This would be true if the drag force were uniformly distributed along the bundle’s height, which is very unlikely considering the no-slip condition at the apical surface of the hair cell. As detailed above (text in blue font), we estimate instead that the correction ought to be closer to 3/2. Thus, from this factor alone, Geleoc et al., might have underestimated the hair-bundle stiffness by a factor 4/3. Finally, note that we measured stiffness 5-10 ms after the onset of step stimuli, which corresponds to the ‘dynamic stiffness’ in Geleoc et al., 1997 (measured in their case 4 ms after the stimulus onset): this ‘dynamic stiffness’ is found to be 5.6 mN/m, thus 25% higher than the ‘steady-state stiffness’ (measured 45 ms after the stimulus onset) of 4.5 mN/m mentioned by the reviewer. With the correction factor 4/3, the stiffness estimates rises to a value of 7.5 mN/m, which is comparable to our current estimate.

In Beurg et al., 2008, the authors used a flexible glass fiber with a silicone bead at its tip to deflect the hair bundle. As pointed out by reviewer 1, we use a different stimulation technique: the fluid jet. With fiber stimulation, only motion at the fiber’s tip is measured; the underlying hair bundle is not visualized. In addition, the force applied to the bundle is somewhat distributed among the different stereocilia; the contact zone between the silicone bead and the bundle is poorly characterized (Nam et al., 2015). It is thus unsure that all the stereocilia are recruited in concert by this stimulation technique. Because OHC bundles are known to be weakly cohesive, stimulation of a subset of stereocilia would result in an underestimate of hair-bundle stiffness. As a matter of fact, finite-element simulations of the hair-bundle response to fiber stimulation suggest that the estimated stiffness in OHCs was 45% smaller than the actual stiffness (Nam et al., 2015). Applying this correction, the stiffness of OHC bundles at the 4-kHz location would be ~7 mN/m. With these caveats in mind, hydrodynamic stimulation may be advantageous. In accordance with a more cohesive motion of the hair bundle and/or tighter coupling between the stimulus and bundle motion in response to a fluid jet, the current-displacement relation resulting from fluid-jet stimulation has been shown to be much narrower (5-95% width: ~100 nm) than that measured with fiber stimulation (5-95% width: ~500 nm; compare Figure 1E to Figure 3C in Johnson et al., (2011)).

1) Impact statement: “The mechanical properties of the tip-link complex contribute to frequency selectivity of auditory hair cells”. See also Discussion section: “tip-link complex plays a role in setting the CF of the hair cell”. To my knowledge there is no evidence to support these contentions. Moreover, the present consensus (see e.g., work by Grosh and colleagues) is that prestin-based somatic contractility is sufficient to account for cochlear amplification.

Although our findings provide no direct proof that the observed tonotopic gradient in stiffness and tension of the tip-link complex plays a role in setting the characteristic frequency of the hair cell, they reveal a striking covariation of the mechanics of the tip-link complex and of the characteristic frequency of the hair cell. Our work thus adds to the available evidence indicating that multiple physical properties of the hair cell are tuned according to the cell’s characteristic frequency, especially in outer hair cells. To the very least, our work opens an avenue for further research and modelling, which ought to determine whether and how stiffness and tension of the tip-link complex influence the characteristic frequency of the corresponding hair cell.

The reviewer implies that there is no need for implicating the tip-link complex in frequency tuning of the hair cell because electromotility is sufficient to account for cochlear amplification. There is no doubt that electromotility is necessary for frequency-selective cochlear amplification but electromotility, by itself, displays a flat frequency response to sinusoidal variations of the transmembrane potential (Frank et al., 1999) and thus provides no frequency tuning. Frequency-selectivity instead results from a complex (and still unclear) interplay between electromotility and its mechanical load. This load includes the hair bundle, whose mechanical properties are determined in part by those of the tip-link complex as shown here (Figure 3) and by others (Hudspeth, (2014)). Notably, there is evidence at the cochlear apex of the Mongolian gerbil that the tip-link complex may contribute significantly to the mechanical impedance of the cochlear partition (Chan and Hudspeth, (2005)). Finally, the work of Karl Grosh’s group in fact acknowledges a potential role of the hair bundle in cochlear amplification, at least at the apical end of the cochlea (see the Discussion section in Meaud and Grosh, (2011)).

Still, the reviewer’s comment made us realize that our significance statement may lead to some misunderstanding. We now propose: “The tip-link complex of the hair cell is mechanically tuned along the tonotopic axis of the cochlea.” In addition, we have toned down our assertion about the significance of our observation for frequency selectivity at the end of the first paragraph of the Discussion section, which now reads: “In return, our data raise the possibility of a mechanical role of the tip-link complex in the process that sets the characteristic frequency of the hair cell”.

2) Discussion section. “Computing the relative difference in stiffness ratio between the two extreme cochlear locations”: are 1 and 4 kHz regarded as extreme locations?

We changed the sentence, which now reads: “Computing the stiffness ratio between the most basal and the most apical cochlear location that we were able to probe, […]”.

3) Discussion section. The section relating stiffness to MT channel number is an intriguing suggestion, but the correlation is poor and not enough of the cochlea is examined. Is it possible there is an error either in determining location or allocating CFs?

We have deleted the paragraph suggesting that the gradient of gating-spring stiffness may be related to the gradient in the number of MT channels.

4) Discussion section. Whether or not the channel gating force is manifest in the bundle's force-displacement plot depends mainly on the bundle's passive stiffness. For example, there is a strong non-linearity for the frog saccular hair bundle, where the passive stiffness is less than 0.5 nmN/m (Cheung and Corey, 2005). For mammalian hair bundles, less stiff vestibular bundles show more non-linearity than more stiff auditory bundles (Geleoc et al., 1997). The present results with stiffness of 12 mN/m would not be expected to show any non-linearity. Please add this.

To address this issue, we have added the following sentences at the end of the first section of the Results section:

“As also observed by others using fluid-jet stimulation of cochlear hair cells (Géléoc et al., 1997; Corns et al., 2014), we measured force-displacement relations that were remarkably linear (Figure 2B and D), showing no sign of gating compliance (Howard and Hudspeth, 1988). There are at least two possible explanations for this observation. First, the rise time (~500 µs; Figure 2−figure supplement 3) of our fluid-jet stimuli may have been too long to outspeed fast adaptation, masking gating compliance (Kennedy et al., 2003; Tinevez et al., 2007). Second, gating forces –the change in tip-link tension evoked by gating of the transduction channels (Markin and Hudspeth, 1995)− may have been too weak to affect hair-bundle mechanics under our experimental conditions (Fettiplace, 2006; Beurg et al., 2008).”

Based on available estimates of the single-channel gating force for cochlear hair cells, we agree that it is not surprising to observe linear force-displacement relations with hair bundles as stiff as those studied here. However, there are reasons to believe that gating forces in mammals may have been underestimated. First, adaptation appears to be so fast in the mammalian cochlea that it is difficult to apply step stimuli that are brisk enough to outspeed adaptation in measurements of current-displacement or force-displacement relations. It has been suggested that strong gating compliance may be masked by fast adaptation at short timescales after the onset of a step stimulus (Tinevez et al., (2007)). Second, the magnitude of the gating force may depend on experimental conditions. With hair cells from the bullfrog sacculus, hair bundles are endowed with only weak gating compliance in a one compartment configuration with standard saline containing 4-mM Ca^2+^ (single-channel gating force of ~0.3 pN in Howard and Hudspeth, (1988)). Instead, one can observe gating compliance that is strong enough to result in negative hair-bundle stiffness in a two-compartment configuration that recapitulates natural conditions (single-channel gating force up to ~0.9 pN in Martin et al., (2000); see also Figure S1 in Tinevez et al., (2007)). With a single-channel gating force of 0.9 pN, the stiffness of an OHC at the 4-kHz location in the rat cochlea would be reduced by 3.3 mN/m as the result of gating compliance, corresponding to nearly 40% of the measured stiffness (using the corrected value 8.6 mN/m): gating forces as high as those measured in frog would thus produce noticeable gating compliance with hair bundles as stiff as those studied here. There is actually a report of a two-compartment preparation of the Mongolian gerbil’s cochlea that showed mechanical nonlinearities at the level of basilar-membrane vibrations that are likely the result of gating forces (Chan and Hudspeth, (2005). Thus, gating compliance could be more visible under circumstances where the endolymphatic/perilymphatic compartments (and perhaps also the endocochlear potential) are preserved. Finally, mechanical stimulation with a fluid jet consistently results in narrower current-displacement relations than stimulation with a glass rod, resulting to larger estimates of the gating forces with the fluid jet (Johnson et al., (2011)). Although these considerations are interesting, we felt that they go beyond the scope of our paper.

Reviewer #2:[…] There is a problematic issue in subsection “The hair-bundle stiffness increases along the tonotopic axis”. It is surprising a first glance that the authors mention neither adaptation and the consequential changes in hair-bundle stiffness, nor gating compliance owing to the opening of transduction channels. This is worrisome inasmuch as the lack of these signatures might imply unhealthy cells. Onlyin subsection “Mechanical stimulation and stiffness measurements” do the authors report the time-dependent changes characteristic of adaptation, which they later demonstrate in Figure 2—figure supplement 4. Moreover, they subsequently mention the period of 5-10 ms after the onset of stimulation at which measurements were made, an interval likely to miss gating compliance owing to the swift adaptation of mammalian hair cells. Note in that context the transient at the outset of the single record with a high temporal resolution (Figure 2—figure supplement 3). Finally, in subsection “Tip-link tension depends on calcium” they state that "gating forces were relatively weak under our experimental conditions," when really their measurement conditions probably precluded the detection of these forces. It would be useful to state at the outset that the principal measurements reflect "steady-state" stiffnesses or the like, and perhaps to note that adaptation was probably observed but was not relevant, and that gating compliance might have been missed for technical reasons.

These are good suggestions and we have amended the text in the subsection “The hair-bundle stiffness increases along the tonotopic axis” accordingly.

First, we have added a paragraph about our observations of the mechanical correlate of adaptation:

“Each hair bundle responded to a force step with a fast deflection in the direction of the stimulus followed by a slower movement in the same direction. Over the duration of the step, the deflection of the hair bundle increased in the direction of the applied step by 22% for inner hair cells and by 12% for outer hair cells, corresponding to an apparent softening of the hair bundle by the same amount (Figure 2‒figure supplement 5). A mechanical creep is expected from mechanical relaxation of tip-link tension associated with myosin-based adaptation (Hudspeth, 2014). Accordingly, the creep was strongly reduced upon tip-link disruption by EDTA treatment.”

Second, as already mentioned in response to reviewer 1’s comments (point 4), we have also added a paragraph discussing gating compliance and the possible reasons that may explain why it was not observed:

“As also observed by others using fluid-jet stimulation of cochlear hair cells (Géléoc et al., 1997; Corns et al., 2014), we measured force-displacement relations that were remarkably linear (Figure 2B and D), showing no sign of gating compliance (Howard and Hudspeth, 1988). There are at least two possible explanations for this observation. First, the rise time (~500 µs; Figure 2−figure supplement 3) of our fluid-jet stimuli may have been too long to outspeed fast adaptation, masking gating compliance (Kennedy et al., 2003; Tinevez et al., 2007). Second, gating forces –the change in tip-link tension evoked by gating of the transduction channels− may have been too weak to affect hair-bundle mechanics under our experimental conditions (Fettiplace, 2006; Beurg et al., 2008).”

Reviewer #3:[…] 1) It is important to be clear that the changes observed with BAPTA treatment are not necessarily limited only to tip-link proteins but anything in series with these proteins. The tonotopic changes described may be a reflection of intrinsic tip link properties but they also can be a reflection of changes described for the TMC proteins, upper insertion point proteins and even the lipid bilayer. The authors do allude to this idea in several places (using the term tip link complex), but it comes across a bit confused and it is unclear when they mean only the tip-link proteins and when they include accessory structures. The discussion of potential intrinsic tip- link protein effects is reasonable, it simply needs to be expanded to be inclusive as there is no clear data one way or the other.

We have expanded the Introduction to clarify that the tip-link complex includes all molecules in series with the tip link, including the transduction channels and the stereociliary membrane, and that the gating spring that controls the stiffness of the tip-link complex could reside anywhere within this protein assembly. We now write:

“Sound evokes hair-bundle deflections, which modulate the extension of elastic elements −the gating springs− connected to mechanosensitive ion channels. […] In the following, the molecular assembly comprising the tip link, the transduction channels, as well as the molecules to which they are mechanically connected will be called the ‘tip-link complex’.”

2) Does the tonotopic change in stereocilia height alter the force sensed from the fluid jet because of the apical surface either reflecting flow or adding a resistance to flow? That is, how sure are the authors that the stimulus to the hair bundle is constant between tonotopic positions?

To estimate the drag force exerted by the fluid jet on a hair bundle, we modeled the hair bundle by a prolate ellipsoid with the long and short axis corresponding to the width (W) and height (h) of the hair bundle, respectively. The expression of the hydrodynamic radius of the ellipsoid (Equation 1) takes into account the height (as well as the width) of the hair bundle; it thus changes along the tonotopic axis of the cochlea. In other words, the force exerted by the fluid jet is expected to vary from one location to the next as the result of tonotopic changes in hair-bundle size, but we accounted for that. In practice, however, the effect is small, nearly negligible (see Figure 3—source data 1). This is not surprising considering that the height of the hair bundle decreases by only 13% and 23% in inner and outer hair cells, respectively, from the apical-to-basal cochlear locations that we probed in our work.

The apical surface of the cochlear tissue should affect the hydrodynamic radius of the hair bundle, whose mathematical expression (Equation 1) is exact only in the case of an ellipsoid immersed in an infinite space of fluid. Because of the no-slip condition at a solid surface and because the bundle’s height is significantly smaller than the typical breadth (noted *A* in Equation 2) of the fluid jet, the surface should impose a cutoff on the fluid-velocity field: the fluid velocity at the bundle’s top ought to be near that along the axis of revolution of the fluid-jet pipette (velocity *V*_max_ in Eq. 2)and drop to zero over the height (h) of the hair bundle. We now acknowledge in the Materials and methods section that the expression given by Equation 1 is approximate: “We note that the hydrodynamic radius given by Equation 1 is exact only for an ellipsoid immersed in an infinite volume of fluid, whereas the hair bundle instead stands erect at the apical surface of the hair cell; this expression is thus an approximation.”

In principle, a change in bundle height results in a change in the correction to the expression of the hydrodynamic radius that we use to calculate the drag force on the hair bundle, potentially affecting the gradients that we measure. However, the height varies by less than 25% in our experiments, whereas the observed mechanical gradients correspond to variations that are much larger (e.g. increase of 240% for the stiffness of outer hair cells between the 1- and 4-kHz locations). Thus, the effect of the surface on the bundle’s hydrodynamic radius as a function of cochlear location should contribute marginally to the observed mechanical gradients. In other words, it is unlikely that a differential effect of the surface on the force applied by the fluid jet contributes significantly to the observed mechanical gradients. The new control experiment that we have added to the manuscript to probe the validity of our force-calibration procedure (in frog; Figure 2—figure supplement 4) also indicates that the approximate expression for the hydrodynamic radius (Equation 1) of the hair bundle provides satisfactory results (see also the added text in the subsection “Applying and measuring forces with the fluid jet”), i.e. that the surface has a weak effect on this expression.

3) Is it possible that the resting tip-link tension variations are a function of tonotopic membrane potential differences operating in a feedback to control driving force and perhaps calcium permeation? Similarly, is it possible that previously described tonotopic differences in calcium buffering might underlie the measured changes in resting tension?

These are interesting suggestions but we felt that they go beyond the scope of the present work. Of course, if tip-link tension were regulated by Ca^2+^ at an intracellular site then we agree that any mechanism that modulates the Ca^2+^ concentration at the regulation site would be relevant, including changes of the transmembrane potential or of Ca^2+^ buffering.

4) Is there concern that the ages used overlap in timing with changes in TMC proteins switching? As development occurs tonotopically, might some of the observations here be due to different developmental states?

The developmental switch between the TMC proteins may indeed be a concern in inner hair cells but not in outer hair cells. In this case, the gradient in tip-link tension (IHCs) may have been slightly overestimated at P8-P10 compared to mature hair cells, underestimating the contrast between the gradients observed in IHCs and OHCs. To address this issue, we have added the following paragraph in the section dealing with the possible effects of hair-cell maturation on our observations:

'Within the tip-link complex, transmembrane channel-like protein isoforms 1 and 2 (TMC1 and TMC2) are thought to be essential components of the transduction channels (Fettiplace and Kim, 2014; Pan et al., 2018). TMC2 is only transiently expressed after birth; TMC1 is expressed later than TMC2 but is fundamental to mechanoelectrical transduction of mature cochlear hair cells (Kawashima et al., 2011; Kim and Fettiplace, 2013). Stereociliary expression levels of TMC1, as well as their tonotopic gradients, were recently shown in mice to be nearly mature by P6, both in inner and outer hair cells, but TMC2 may still be present in apical inner hair cells until about P13 (Beurg et al., 2018). Because TMC2 confers larger Ca^2+^ permeability to the transduction channels (Kim and Fettiplace, 2013), the Ca^2+^ influx at rest in the inner hair cells at P8-P10 may have been larger than at more mature developmental ages, possibly lowering tip-link tension (Figure 6) and steepening the tension gradient.”

5) In subsection “Tip-link tension depends on calcium” the authors appear to be making a statement about adaptation and calcium effects. If they want to go this way, then they should be as quantitative here as they are in the rest of the manuscript. As the data presented here is consistent with that from the Peng papers where calcium dependent and independent effects are described, the calcium dependent effects are simply argued to be separate from adaptation, just as the upper tip link insertion motors are argued to be a tensioning mechanism independent of adaptation. Seems to me that the data presented in this manuscript does not actually directly address the topic of calcium and adaptation. The authors should either make a statement that includes a conclusion, present data on the topic, or eliminate the paragraph.

We have deleted the paragraph. The issue is mostly semantic. As we write in subsection “Tip-link tension depends on calcium”, the effects of Ca^2+^ on tip-link tension (Figure 6) are actually consistent with the mechanical correlate of myosin-based ‘slow’ adaptation, i.e. Ca^2+^-dependent myosin movements up and down the stereocilia to regulate tip-link tension. Instead, as far as we understand, the controversy is about ‘fast’ adaptation, i.e. whether this form of adaptation represents Ca^2+^-dependent channel reclosure or Ca^2+^-independent repacking of membrane lipids near the transduction channels. Peng et al. have proposed to exclude the tensioning mechanism that sets the open probability of the transduction channels from adaptation (Peng et al., 2016), whereas this mechanism was earlier associated with slow adaptation (Hudspeth and Gillespie, 1994; Hudspeth, 2014). Our observation that tip-link tension depends on Ca^2+^ (Figure 6) relates to some observations in Peng et al., 2016 and we have added a paragraph to clarify how (subsection “Tip-tension depends on calcium”):

“a negative deflection would also be produced if lowering the extracellular Ca^2+^ concentration evoked stiffening (Martin et al., 2003; Beurg et al., 2008) or shortening of the gating springs, increasing tension in the tip links. […] Our recordings (Figure 4B and Figure 6D-E) are consistent with this hypothesis but a change in membrane mechanics is unlikely to explain the observed steady-state increase in tip-link tension (Figure 6E) if slow adaptation happens in these cells (Figure2—figure supplement 5).”

6) The last sentence of the manuscript implies that the tonotopic variations in the hair bundle mechanics support an argument that the hair bundle is part of the active process. It seems to me that the data in the manuscript argue that OHCs, which need to work cycle by cycle are mechanically different tonotopically and also from IHCs which do not need to operate cycle by cycle. The data supports an argument for the hair bundle providing some frequency selectivity, perhaps a tuning mechanism, which may or may not be active. It might be more valuable for the authors to better delineate what they are referring to when they say active process, do they mean amplification, do they mean tuning, do they mean both? Seems like a very loaded sentence.

Cochlear amplification is viewed as a frequency-selective ‘active process’, because it cannot be reduced to one of its components, including somatic electromotility, basilar-membrane or hair-bundle mechanics. For instance, although we know since the pioneering work of Georg von Bekesy that the basilar membrane is endowed with a passive resonant behavior, the characteristic frequency of a local section of the (healthy) cochlear partition can be higher or lower than the local (passive) resonance frequency of the basilar membrane that can be assayed *post mortem*. Amplification and frequency selectivity appear instead as emergent properties of the dynamic interplay between all the vibrating components of the cochlear partition (Hudspeth et al., 2010; Ó Maoiléidigh and Jülicher, 2010; Meaud and Grosh, 2011; Hudspeth, 2014). One cannot really separate amplification from frequency tuning, for amplification is frequency selective and thus provides active filtering of sound inputs.

We have expanded the conclusion of the manuscript along these lines and now write: “Thus, the division of labor between inner and outer hair cells may impart more stringent regulatory constrains to outer hair cells to tune their mechanoreceptive antenna according to the local characteristic frequency of the cochlear partition. However, the exact contribution of the hair bundle to frequency tuning remains unsure and, more generally, the mechanism that specifies the characteristic frequency remains a fundamental problem in auditory physiology. This may be in part because frequency selectivity cannot be ascribed to one element only, for instance the passive resonant property of the basilar membrane that was characterized in the pioneering work of von Bekesy (Von Békésy and Wever, 1960). Various models of cochlear mechanics instead indicate that the characteristic frequency emerges from an active dynamic interplay between somatic electromotility of outer hair cells (Ashmore, 2008) and the micromechanical environment, including the basilar and tectorial membranes, as well as the hair bundle (Nobili and Mammano, 1996; Hudspeth et al., 2010; Ó Maoiléidigh and Jülicher, 2010; Meaud and Grosh, 2011; Hudspeth, 2014; Reichenbach and Hudspeth, 2014). Mechanical tuning of the inner constituents of the cochlear partition appears to happen at many scales: from the mesoscopic scale of the basilar and tectorial membranes, to the cellular scale of the hair bundle and hair-cell soma, down to the molecular scale of the hair cell’s transduction apparatus. Our work demonstrates that tonotopy is associated, in addition to other factors, with stiffness and tension gradients of the tip-link complex.

[Editors' note: further revisions were requested prior to acceptance, as described below.]

However, two of the reviewers remain concerned about the claims relating to "tonotopy" in the manuscript. We accept the reasons given for not being able to obtain responses at the 7 kHz location, but this does mean that only 20% of the length of the cochlea was assessed. We therefore think it would be prudent to stress this in the paper to avoid giving a misleading impression about the range of frequency locations that were sampled.

Thank you for evaluating our manuscript favorably. To address the remaining concern of two of the reviewers, we have further emphasized throughout the text of our revised manuscript that we explored only part of the tonotopic axis. Note that our work deals with both inner and outer hair cells and that the range of positions covered for inner hair cells (50% of the tonotopic axis; CF 1-15 kHz) was larger than that for outer hair cells (20% of the tonotopic axis; CF 1-4 kHz). With the changes listed below, we believe that we can’t give a misleading impression to the reader. It is clear that future work ought to determine whether or not the mechanical gradients that we reveal here within the apical half of the cochlea extend to a larger portion of the tonotopic axis. With the data and techniques at hand, it is currently difficult to argue one way or another.

List of changes:

Abstract: “[…] we have investigated the micromechanical properties of the hair cell’s mechanoreceptive hair bundle within the apical half of the rat cochlea.”

End of the Introduction: “In this work, we probed passive and active hair-bundle mechanics along the tonotopic axis of an excised preparation of the rat cochlea, within an apical region dedicated to the detection of relatively low sound frequencies for this animal species (1‒15 kHz; Figure 1).”

Legend of Figure 1: “Recordings were made at locations marked by black disks, corresponding to characteristic frequencies (in kHz) increasing from the apex to the base of the cochlea as indicated on the figure and to fractional distances from the apex of 5%, 10%, 20%, and 50%. We report measurements from both inner and outer hair cells at the 1‒4-kHz locations, but only from inner hair cells at the 15-kHz location.”